# A smart pathogen detector engineered from intracellular hydrogelation of DNA-decorated macrophages

Yueyue Gui[1,4], Yujing Zeng[2,4], Binrui Chen[1], Yueping Yang[1], Jiehua Ma[3] & Chao Li [1]✉

Bacterial infection is a major threat to global public health, which urgently requires useful tools to rapidly analyze pathogens in the early stages of infection. Herein, we develop a smart macrophage (Mø)-based bacteria detector, which can recognize, capture, enrich and detect different bacteria and their secreted exotoxins. We transform the fragile native Møs into robust gelated cell particles (GMøs) using photo-activated crosslinking chemistry, which retains membrane integrity and recognition capacity for different microbes. Meanwhile, these GMøs equipped with magnetic nanoparticles and DNA sensing elements can not only respond to an external magnet for facile bacteria collection, but allow the detection of multiple types of bacteria in a single assay. Additionally, we design a propidium iodide-based staining assay to rapidly detect pathogen-associated exotoxins at ultralow concentrations. Overall, these nanoengineered cell particles have broad applicability in the analysis of bacteria, and could potentially be used for the management and diagnosis of infectious diseases.

The pathogenic infection is a leading cause of death worldwide, which causes epidemics and huge public expenditure[1,2]. Recently, the evolution of bacteria has been greatly promoted because of the misuse and overuse of antibiotics. Consequently, the emergence of various drug-resistant strains, such as methicillin-resistant *Staphylococcus aureus*, multidrug-resistant *Streptococcus pneumoniae* and multidrug-resistant *Pseudomonas aeruginosa* and so forth, has become a severe problem for clinical acquired infections[3–5]. The *AMR Review* published in the United Kingdom estimated that about 700,000 people worldwide die of drug-resistant bacteria infection every year, and this number will increase to 10 million by 2050, with an estimated economic loss of 100 trillion dollars[6]. Therefore, approaches to efficiently identify pathogens are urgently needed to control bacterial transmission and facilitate anti-infection therapy.

Currently, plate culture is widely used for bacterial identification; however, the main drawback of this method lies in its long analysis time (usually more than 48 h) and the requirement of specialized culture conditions[7], which limits its speed and availability. Nucleic acid tests (e.g., polymerase chain reaction, PCR) using nucleic acid primers have excellent sensitivity, but these tests suffer from a complicated workflow (e.g., cell lysis, nucleic acid extraction, magnetic separation, washing and amplification, etc.) and require expensive protein enzymes and thermal cyclers[8–10]. Alternatively, immunological tests such as lateral flow strip-based assay are an attractive platform for detecting pathogens because of their simplicity, low cost, and rapid signal generation[11,12]. However, frequently, a pair of high-quality bioreceptors (e.g., antibodies or aptamers) must be screened and optimized for both recognizing and sensing in a sandwich-type immunoassay, which restricts the sensor design and deployment. Meanwhile, the information on the targeted bacteria should be known in advance. When the pathogens are unknown, it is difficult to choose appropriate bioreceptors for bacteria capture and detection[13]. Besides

[1]School of Food and Biological Engineering, Hefei University of Technology, 230009 Hefei, P. R. China. [2]State Key Laboratory of Analytical Chemistry for Life Science, School of Life Sciences, Nanjing University, 210023 Nanjing, P. R. China. [3]Tongren Hospital, Shanghai Jiao Tong University School of Medicine, 200336 Shanghai, P. R. China. [4]These authors contributed equally: Yueyue Gui, Yujing Zeng. ✉e-mail: lchao@hfut.edu.cn

cell analysis, pathogens often secrete various exotoxins (e.g., hemolysin) as their weapons to kill the target host cells, which can be used as biomarkers to monitor the virulence of bacteria[14]. However, approaches that can simultaneously analyze both bacterial cells and their virulence factors are still rarely explored.

Over millions of years of evolution, innate immune cells such as macrophages (Møs), dendritic cells, and neutrophils have optimized an incredible recognition capacity for a broad-spectrum of pathogens and their virulence factors[15]. Typically, these immune cells can't recognize a certain bacterium but rely on a set of surface receptors such as toll-like receptors (TLRs)[16], mannose receptors[17], scavenger receptors[18], and complement receptors[19] and so forth to bind given components of pathogens (known as pathogen-associated molecular patterns, PAMPs)[20]. Among them, mannose receptor (MR), a 180 kDa transmembrane protein, is able to recognize the patterns of carbohydrates on the infectious agents such as Gram-positive and Gram-negative bacteria, yeasts, and parasites[21]. When immune cells are stimulated by cytokines, pathogens and their products, the upregulation of MR can significantly enhance their recognition and phagocytosis capacity for battling infection[22,23]. In view of the strong correlation between immune cells and pathogens, we envision immune cells as a useful tool for pathogen analysis. However, developing immune cells as a smart detector that can efficiently recognize, enrich, and report specific pathogen remains a big challenge because of the following reasons. First, living Møs have low mechanical stability, which is fragile to various operations and easily damaged, making it difficult to practical use. Second, cells are not facilely manipulated, which frequently involves multiple centrifugation and washing steps, making them inconvenient for routine applications. Last but not least, immune cells can't distinguish specific bacteria species, let alone produce detectable signals even though they recognize and bind pathogens.

Herein, we report a facile strategy to transform living Møs into a smart bacteria detector by combining the intracellular hydrogelation technique and DNA nanotechnology. Very recently, Hu and coworkers described a facile hydrogelation approach to assemble synthetic hydrogels inside the cells without disturbing the fluid and functional plasma membranes, making them suitable for subsequent biological application (e.g., ex vivo T cell modulation)[24,25]. In this work, we apply this intracellular hydrogelation technique to transform Møs and the resulting gelated Møs (GMøs) can successfully overcome the abovementioned problems, making them suitable for in vitro use. With this tool in hand, we can achieve several goals: (1) GMøs have robust gelated cores and intact cell membranes, which allows them to resist different harsh conditions destructive to living cells. (2) GMøs pretreated with magnetic nanoparticles (MNPs) can be manipulated by an external magnet, so we can use an external magnet to conveniently separate bacteria-adsorbed cell particles from complex biological media with minimal loss. Since the existence of a large number of impurities in the biological samples, magnetic separation plays a fundamental role in the following analysis. (3) GMøs can efficiently recognize, capture, and enrich a broad spectrum of bacteria to their surface, and the close proximity distance between bacteria and the underlying cell membrane will easily activate sensing elements or facilitate the adsorption of bacteria-associated secretions on the cell membrane, which can dramatically improve the detection performance. (4) GMøs can be decorated with responsive DNA devices (e.g., DNAzyme) for specific bacteria biosensing using a capture-and-detect strategy. Moreover, because of the structural characteristics of GMøs, they can be further applied to analyze bacteria-associated toxins, which not only provides more information about the virulence of the infected bacteria but also further extends the applicability of the GMø-based assay. To the best of our knowledge, exploring immune cells as a smart detector for pathogen analysis in vitro has not been described, which may provide a clue to rationally design next-generation cell-based biosensors for fighting against bacteria.

## Results
### Preparation of DNA-GMøs
The preparation procedure of DNA-decorated, gelated macrophages (DNA-GMøs) is shown in Fig. 1. Briefly, Møs are first incubated with citrate-coated superparamagnetic iron oxide nanoparticles (MNPs, diameter ≈ 100 nm, Supplementary Fig. 1) to obtain magnetic cells because of the efficient internalization of MNPs. In parallel, these cells are stimulated by a cytokine

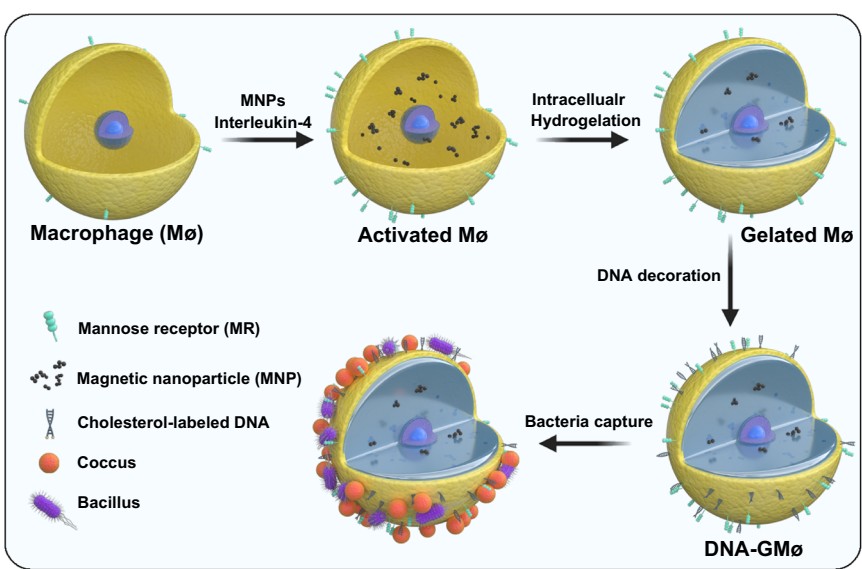

**Fig. 1 | Schematic illustration of the preparation of DNA-GMøs for bacteria recognition and capture.** Activated Møs with magnetic response and upregulated protein receptors were obtained by treating the cells with MNPs and interleukin-4. Then, the Møs were transformed into gelated cell particles (GMøs) using an intracellular hydrogelation method. Finally, the membrane of GMøs was decorated with DNA sensing elements, and the resulting DNA-GMøs can be used to efficiently capture and detect different bacteria.

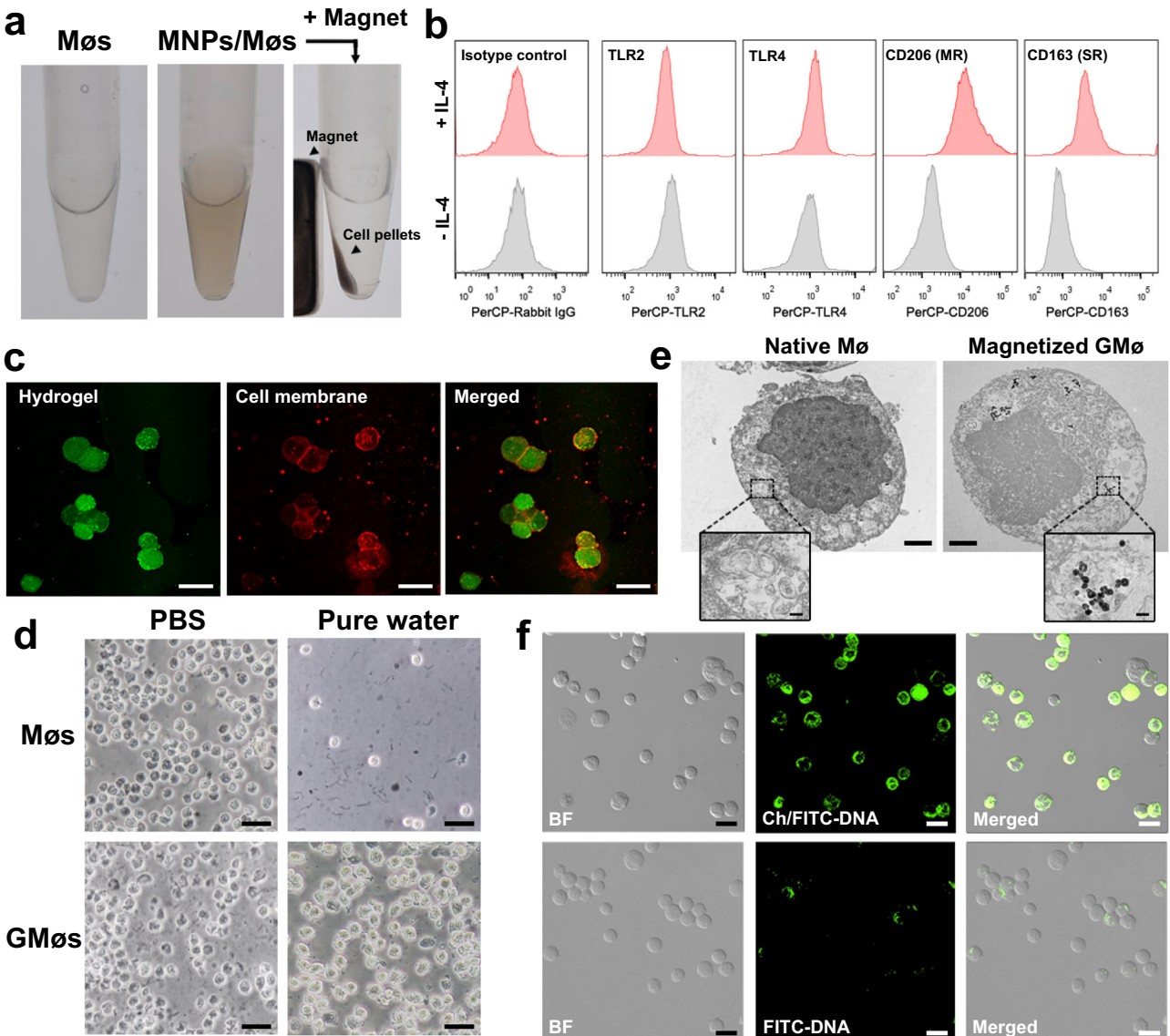

**Fig. 2 | Preparation and characterization of DNA-GMøs. a** Photographs of cell pellets before and after incubation with MNPs and the response of the magnetized cells to an external magnet. **b** Flow cytometry analysis of TLR2, TLR4, CD206 (MR), and CD163 (SR) expressions in Mø before and after stimulation of IL-4. Rabbit IgG antibody is used as an isotype control. **c** Confocal images of the GMøs with fluorescein-diacrylate (green) for hydrogel labeling and DiI dye (red) for membrane staining. Scale bars = 20 μm. **d** Bright-field microscopy of Møs and GMøs upon suspension in PBS and pure water. Scale bar: 30 μm. **e** TEM images show the cross-sectional structure of native and magnetized GMøs. Scale bar is 2 μm. Insert picture shows a higher magnification view of phagosomes in the cells. Scale bar is 100 nm. **f** Confocal images of GMøs incubated with the fluorescein isothiocyanate-labeled DNA with (Ch/FITC-DNA) or without cholesterols (FITC-DNA). BF represents bright field. DNA molecules are shown in green. Scale bar is 20 μm.

(e.g., interleukin-4, IL-4) to induce the upregulation of pathogen-binding receptors (e.g., MR and CD163). Of note, we use RAW264.7 (a murine macrophage cell line) which is one of the most widely used immortalized macrophages for in vitro experiments as the basic cell material because of its convenient large-scale production and easy maintenance without adding any inducers. Then, the intracellular hydrogelation is achieved by direct permeation of photoinitiators (2-hydroxyl-4'-(2-hydroxyethoxy)-2-methylpropiophenone, I2959) and monomers (poly(ethylene glycol) diacrylate, PEG-DA, $M_w$ = 700 Da) and subsequent UV irradiation according to previous reports[24]. Finally, DNA sensing elements (e.g., DNAzyme) responding to specific bacteria can be facilely decorated on the gelated cells for convenient, rapid, and highly sensitive fluorescence readout. Consequently, magnetic DNA-GMøs that can recognize, enrich, and detect multiple types of microorganisms are produced.

## Characterization of the fabricated DNA-GMøs

We used different techniques to confirm the successful preparation of DNA-GMøs. As shown in Fig. 2a, the resulting cells (MNPs/Møs) became dark compared to natural Møs and could rapidly respond to an external magnet with a high cell recovery (≈98%). Moreover, UV/vis spectra indicated the presence of characteristic peak of MNPs for the MNPs/GMøs sample (Supplementary Fig. 2), further suggesting the successful magnetization of the Møs. Of note, the uptake of MNPs did not affect the viability of Møs (Supplementary Fig. 3), and we also measured intracellular iron contents by inductively coupled plasma mass spectrometry (ICP-MS) to quantitatively assess the cellular uptake of the MNPs at different time points (Supplementary Fig. 4). After treatment with IL-4, M0-type Møs are transformed into M2-type Møs with upregulation of some unique markers such as CD206, CD163, Arginase-1 (Arg1), Fizz1 and so on[26]. Among them, only CD163 (i.e., scavenger receptor, SR) and CD206 (i.e., mannose receptor, MR) exist

on the cell membrane and directly participate in bacteria recognition and binding[17,27], so their expression on the membrane was evaluated using flow cytometry analysis. As a comparison, other potential markers such as TLR2 and TLR4 that can bind to bacterial components were also tested. The expression of CD163 and CD206 but not TLR2 and TLR4 was significantly promoted using IL-4 stimulation (Fig. 2b and Supplementary Fig. 5), consistent with previous reports[22,28]. The up-regulation of CD206 and CD163 implied that they could have more chances to capture bacteria compared with TLR receptors. Subsequently, the formation of hydrogel networks in the intracellular region was confirmed by fluorescence microscopy. The hydrogel network was stained with green fluorescence by introduction of fluorescein-diacrylate into the cross-linker mixture, and the cell membrane was stained with a lipophilic DiI fluorophore with red fluorescence. The resultant GMøs exhibited discriminative membranous and hydrogel parts (Fig. 2c), suggesting the successful intracellular hydrogelation, which is in well agreement with the previous results[25]. Also, the water exposure assay could be used to rapidly evaluate the hydrogelation of Møs since living Møs rapidly ruptured under hypo-osmotic stress[25], whereas GMøs with robust crosslinked interiors remained intact without obvious morphology change (Fig. 2d and Supplementary Fig. 6). Transmission electron microscopy (TEM) results demonstrated that GMøs had a perforated, hydrogel-filled interior compared with natural Møs (Fig. 2e). Meanwhile, many MNPs were also observed in the phagocytic vesicles (Fig. 2e, inset), further verifying that their magnetic response resulted from the internalization of MNPs. Also of note, these gelated Møs stored in the PBS solution rapidly died without nutrient supply (Supplementary Fig. 7); however, since the cytoplasm of GMøs is solidified by the network of hydrogel, the resulting cell particles retained their membrane integrity and membrane fluidity (Supplementary Fig. 8), suggesting that the protein receptors on the membrane could preserve their functions.

To facilitate the DNA modification, we used cholesterol-labeled DNA duplex (Ch-DNA, Supplementary Table 1), which could spontaneously insert into the cell membrane via strong hydrophobic interactions. To demonstrate this, we conjugated the DNA strands with a fluorophore (i.e., FITC). GMøs were incubated in the presence of Ch/FITC-DNA (100 nM) for 15 min, followed by confocal imaging. DNA strands without cholesterol were used as the control group. As shown in Fig. 2f, GMøs treated with Ch/FITC-DNA exhibited green fluorescence on the cell membrane; by contrast, cells incubated with cholesterol-free DNA strands did not show any fluorescence, thus suggesting that the successful surface immobilization of DNA molecules was due to cholesterol insertion. The average density of DNA was estimated to be $2.4 \times 10^5$ molecules/cell according to the change in fluorescence spectra (Supplementary Fig. 9).

## Stability of the fabricated DNA-GMøs

To evaluate the stability of the prepared DNA-GMøs, we challenged the cells with different conditions, such as hypertonic stress, repeated freeze–thaw cycles, sonication, high-speed centrifugation, and long-time storage (30 days). For a comparison, we used living Møs (LMøs) and paraformaldehyde-fixed Møs (FMøs) as controls. As shown in Fig. 3a, DNA-LMøs were rather fragile, and all treatments caused their rapid deformation (e.g., hypertonic stress), aggregation (e.g., centrifugation) or fragmentation (e.g., sonication), making them impossible for routine experiments in vitro. Although DNA-FMøs had better stability compared to DNA-LMøs, they were still susceptible to freeze–thaw cycles and sonication, which destroyed the majority of cells. Impressively, DNA-GMøs survived under all conditions, evidenced by the negligible change in cell morphology and number (Fig. 3b). Overall, these results confirm the high stability of the DNA-GMøs, thus facilitating their downstream applications such as pathogen capture and detection.

In view of the important role of DNA molecules modified on the GMøs in bacteria detection (*vide infra*), their stability on the cell membrane was also evaluated. Since the cells were gelated, the anchored DNA strands were not internalized into the cells, which could be displaced on the membrane for a long period without obvious loss, as evidenced by fluorescence intensity change determined by flow cytometry and confocal analysis (2 weeks, Fig. 3c). Previous reports revealed that single cholesterol-labeled DNA strands are prone to detach from the lipid membrane in a protein-rich environment[29], so the retention time of DNA strands on the DNA-GMøs dispersed in complex biological media (i.e., 10% serum) was investigated using flow cytometry. As shown in Fig. 3d, the fluorescence of dual-cholesterol-labeled DNA-modified GMøs remained unchanged, implying the negligible detachment of DNA strands from GMøs. This improved stability was attributed to the enhanced affinity of dual cholesterols toward the cell membrane, which is important for subsequent bioapplications.

## Recognition and capture capacity of GMøs for different microbes

Next, we examined the performance of GMøs for binding and isolation of pathogens using two representative bacteria, i.e., Gram-negative *Escherichia coli* (*E. coli*) and Gram-positive *Staphylococcus aureus* (*S. aureus*) as model pathogens. After incubating the GMøs with bacteria suspension for 15 min, the cell particles were collected using a magnet for microscope observation (Fig. 4a). Obviously, the capture capacity of the IL-4-stimulated GMøs was much better than that of unstimulated GMøs for both species (Fig. 4a). The isolated bacteria were alive (Supplementary Fig. 10), which could be analyzed by plate assay. Moreover, we also prepared another gelated cell particle using MCF-7 cells (human breast cancer cell line), which is not an immune cell. As shown in Supplementary Fig. 11, the MCF-7-based particles had a weak binding tendency toward bacteria, which was significantly lower than that of GMøs, thus emphasizing the role of immune cells as the scaffold material. The colonies of *E. coli* and *S. aureus* captured by IL-4-stimulated GMøs were 4.3-fold and 5.5-fold higher than those of unstimulated GMøs (Fig. 4b, c), respectively, consistent with the microscopy results. Titration assay further verified the advantage of cytokine stimulation for bacteria capture (Supplementary Fig. 12). Also of note, when the protein receptors (i.e., MR and SR) on the GMøs were simultaneously blocked by corresponding antibodies, their capture efficiency for bacteria was obviously inhibited (Supplementary Fig. 13), thereby emphasizing the important role of protein receptors in bacteria recognition and capture. Of note, the use of blocking antibodies could not completely inhibit the binding of bacteria to GMøs, highlighting that other receptors are likely involved. Furthermore, the bacteria capture efficiency of the stimulated GMøs slightly reduced after long-term storage (e.g., 1 month), which was beneficial to their real applications. This was probably due to the unchanged protein markers (e.g., MR and SR) on the cell membrane (Supplementary Fig. 14). Therefore, the stimulated GMøs were used for subsequent bacteria analysis.

Figure 4d confirmed that the GMøs could be used to capture other microorganisms, including *Pseudomonas aeruginosa* (*P. aeruginosa*), *Vibrio parahemolyticus* (*V. parahemolyticus*), *Salmonella enteritidis* (*S. enteritidis*) and *Candida albicans* (*C. albicans*). When three bacteria (*E. coli*, *S. aureus*, and *P. aeruginosa*) were mixed in one sample, GMøs could simultaneously capture all species, thus proving the broad-spectrum capture capacity of GMøs (Supplementary Fig. 15). As a comparison, commercially available antibody-modified magnetic beads (Ab-MBs) only

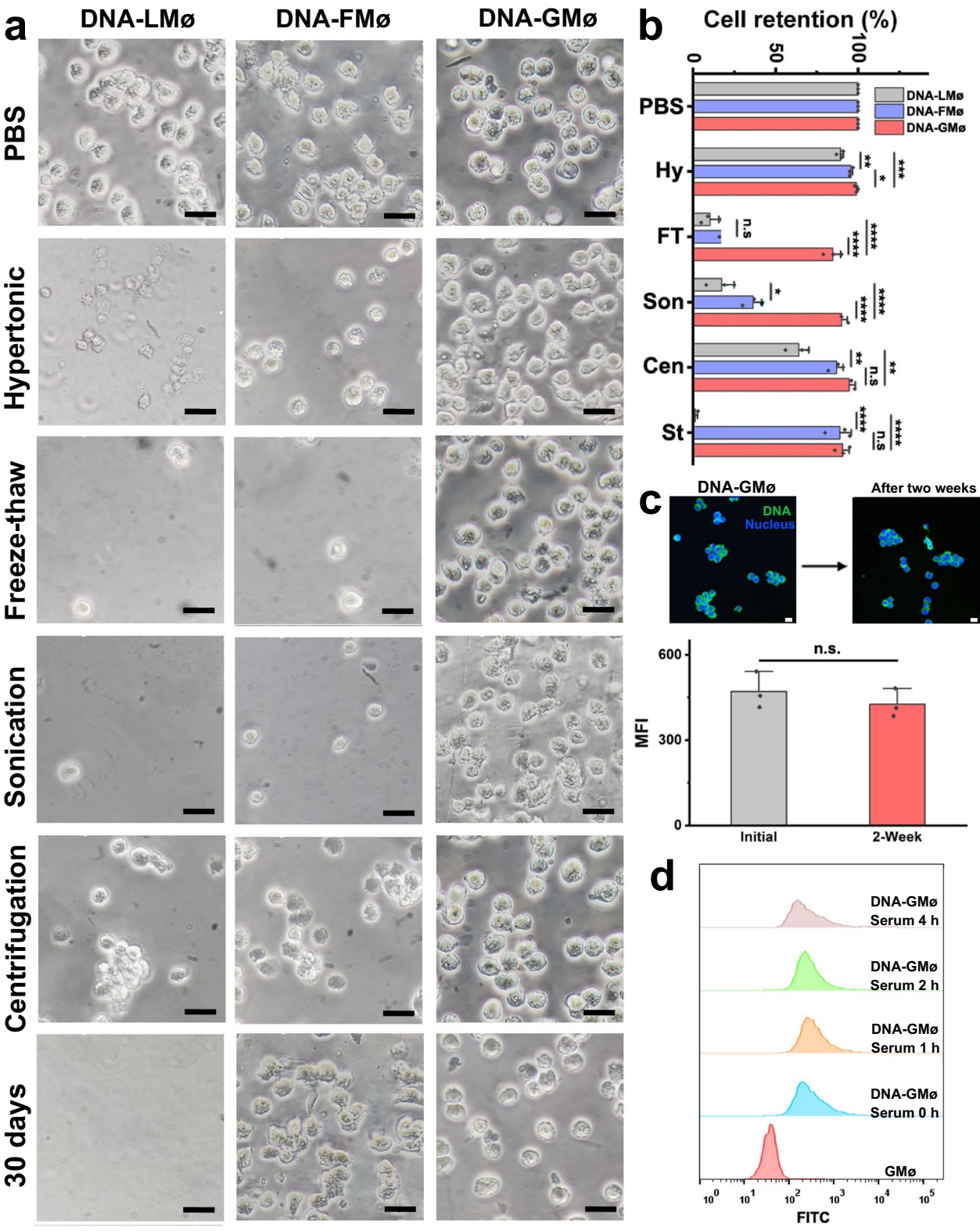

captured the corresponding bacteria due to the specific binding between antibodies and antigens on the bacteria (Supplementary Fig. 16). Moreover, although the size of Ab-MBs (diameter = 10 µm) was compatible to that of GMøs, their capture capacity was significantly lower than that of GMøs. The performance of these particles for bacteria collection from complex media such as plasma, serum, saliva, and urine was also investigated using plate assay. Different from conventional Ab-MBs, the capture capacity of

GMøs was not influenced by complex media (Fig. 4e), further emphasizing their advantages since biological media contain numerous impurities that may affect molecular recognition.

It was not surprising that the performance of Ab-MBs was affected by many factors, such as antibody quality, modification quantity, surface fouling, and correct orientation of Abs on the beads, which all contributed to their limited capture efficiency[30]. In contrast, cell particles had a natural antifouling capacity and

**Fig. 3 | The stability of DNA-GMøs under environmental conditions. a** Bright-field microscopy of DNA-modified living Møs (DNA-LMøs), DNA-modified fixed Møs (DNA-FMøs), and DNA-modified gelated Møs (DNA-GMøs) with different treatments, including hypertonic stress (1 M NaCl), freeze–thaw cycles (3 times), sonication, high-speed centrifugation (10600 g), and long-term storage (30 days). Cells stored in phosphate buffer saline (PBS) solution were used as a control. Scale bar: 20 μm. **b** Statistical analysis of the cell numbers after different treatments. Hypertonic treatment (Hy, 1 M NaCl, 15 min), freeze−thaw cycles (FT, three times), sonication (Son, 40 W, 10 min), centrifugation (Cen, 10,600 × g, 10 min, four times), and long-term storage (St, 30 days). The error bars represent mean ± SEM, n = 3.

Statistical analysis was performed using one-way analysis of variance (ANOVA), *P < 0.05; **P < 0.01; ***P < 0.001; ****P < 0.0001; n.s. not significant. **c** Fluorescence intensity of DNA-GMøs determined by flow cytometry analysis before and after 2 weeks of storage. Insets show the confocal images of DNA-GMøs. Nuclei are shown in blue and DNA molecules are shown in green. The error bars represent mean ± SEM, n = 3. Statistical analysis was performed using two-sided paired-sample t test. n.s.: not significant. Scale bar: 20 μm. **d** Flow cytometry analysis demonstrating the stability of DNA molecules on the GMøs after incubation with 10% bovine serum for different time points. Source data are provided as a Source data file.

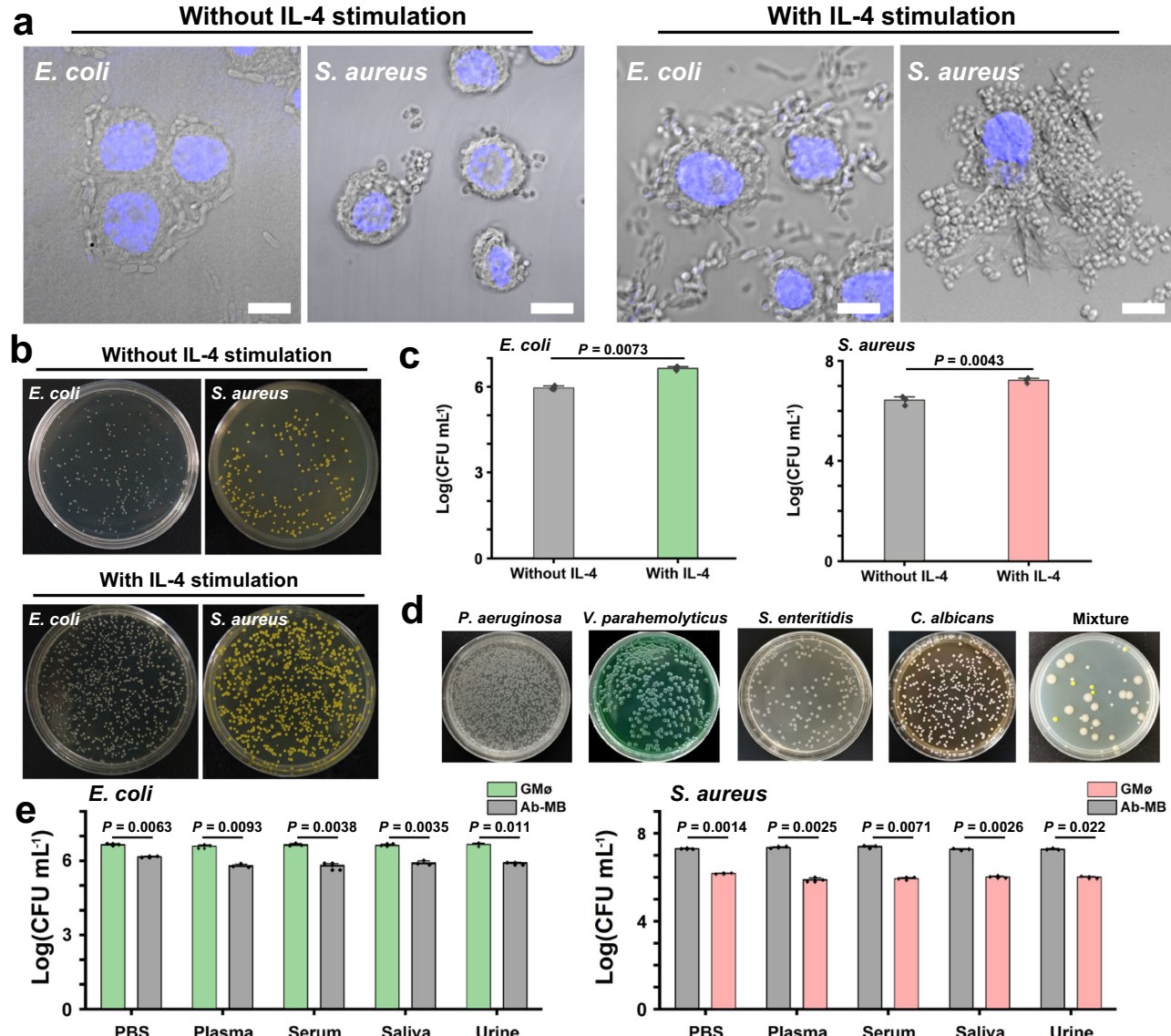

**Fig. 4 | Investigation of GMø-based bacteria recognition and capture.**
**a** Confocal images of GMøs with or without IL-4 stimulation after incubation with *E. coli* (EC) and *S. aureus* (SA). Scale bars: 10 μm. Nuclei are shown in blue. **b** Images of the colonies formed on Luria-Bertani (LB) broth-agar plates captured by GMøs with or without IL-4 stimulation. **c** Statistical analysis of the captured bacteria by two GMøs using a plate assay. Statistical analysis was performed using two-sample t

test. **d** Capture capacity of the GMøs for different bacteria and their mixture. The mixture contains three bacteria, i.e., *E. coli* (red arrow), *S. aureus* (blue arrow), and *P. aeruginosa* (green arrow). **e** The capture performance of the stimulated GMøs and Ab-MBs for *E. coli* (left panel) and *S. aureus* (right panel) in complex biological media. The error bars represent mean ± SEM, n = 3. Statistical analysis was performed using two-sample t test. Source data are provided as a Source data file.

the protein receptors could faithfully fulfill functions as long as the membrane integrity was preserved. Finally, we also confirmed that the modification of DNA molecules on the GMøs did not affect their capture capacity (Supplementary Fig. 17), which was important for subsequent bacteria analysis.

## Specific detection of captured bacteria using DNAzyme-decorated GMøs

Having confirmed the broad capture efficacy of the GMøs, we explored the use of these particles for the specific detection of pathogens. As a proof of concept, we modified *E. coli*-targeting DNAzymes on the

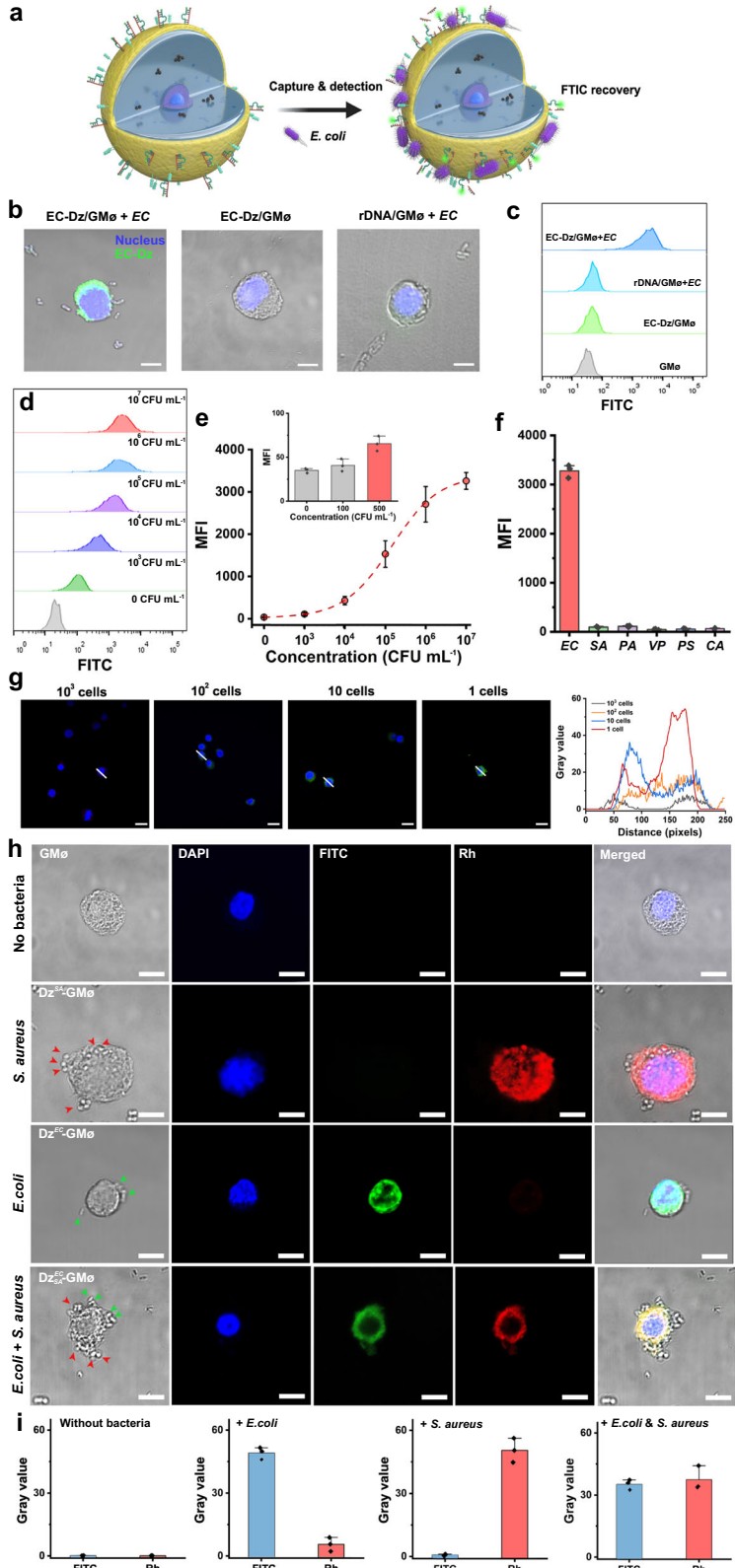

**Fig. 5 | Specific detection of captured bacteria using Dz-GMøs. a** Schematic illustration of the sensing principle of the GMø-based bacteria assay. **b** Confocal images and (**c**) flow cytometry analysis of different GMøs in the absence or presence of *E. coli*. Scale bars: 10 μm. **d** Flow cytometry results of Dz$^{EC}$-GMø-based assay for the detection of *E. coli* with varying concentrations from 0 to 10$^7$ CFU mL$^{-1}$. **e** Fluorescence intensity of the method determining different concentrations of *E. coli*. MFI: mean fluorescence intensity. Inset shows the fluorescence intensity with low *E. coli* concentrations. **f** Specificity study of Dz$^{EC}$-GMøs for *E. coli* detection. *P. aeruginosa*: *PA*, *V. parahemolyticus*: *VP*, *S. pyogenes*: SP, and *C. albicans*: *CA*. **g** A single Dz$^{EC}$-GMø-based assay allows ultrasensitive detection of *E. coli*. **h** Multiplex detection of *E. coli* and *S. aureus* using a Dz$^{EC}_{SA}$-GMø-based assay. Nucleus is shown in blue, activated Dz$^{EC}$ is shown in green (fluorescein Isothiocyanate, FITC), and activated Dz$^{SA}$ is shown in red (rhodamine, Rh). Scale bars: 10 μm. Nuclei are stained by 4,6-diamino-2-phenyl indole (DAPI) and are shown in blue. **i** Fluorescence profiling of the Dz$^{EC}_{SA}$-GMøs after incubating with different bacteria. The error bars represent mean ± SEM, *n* = 3. Source data are provided as a Source data file.

GMøs (termed Dz$^{EC}$-GMøs), which could be used to detect living *E. coli*. Dz$^{EC}$ is selected by Li and coworkers[31–33], which cleaves a fluorogenic DNA/RNA substrate in the presence of crude extracellular mixtures (CEMs) secreted by bacteria.

The sensing principle is shown in Fig. 5a. Specifically, both Dz$^{EC}$ and substrate were immobilized on the cell membrane via a cholesterol insertion method, and the substrate had a FITC fluorophore and BHQ1 quencher. In the absence of *E. coli*, the hybridization between DNAzymes and substrates results in the quenching of the fluorescence of FITC. Nevertheless, in the presence of analyte, the DNAzymes were activated by living bacteria, resulting in the cleavage of the substrate and the recovery of FITC. Because of the close proximity between *E. coli* and DNA sensing elements on the cell membrane, we envision that the CEMs released from bacteria will more efficiently activate DNAzymes than traditional DNAzyme-based approaches, thereby bringing better detection performance.

Figure 5b shows negligible fluorescence on the Dz$^{EC}$-GMøs in the absence of *E. coli*, suggesting the low background signal of this cell-based sensor. However, once bacteria were captured, strong green fluorescence was observed around the cell membrane, suggesting the occurrence of cleavage reactions. To further demonstrate this, we replaced the DNAzyme with a random DNA strand (rDNA), and the resulting rDNA-GMøs failed to produce any signal even after the addition of *E. coli*. To avoid observation bias, flow cytometry was also used to analyze the average fluorescence change of the samples, and a similar trend was observed for the Dz$^{EC}$-GMøs and rDNA-GMøs in the presence of *E. coli* (Fig. 5c), further confirming that the fluorescence was ascribed to the activation of DNAzymes. Additionally, the performance of this method is not compromised by using complex biological samples such as serum, saliva, and urine (Supplementary Fig. 18).

Next, flow cytometry was used to quantitatively detect *E. coli* with a concentration ranging from $10^3$ to $10^7$ CFU mL$^{-1}$ (Fig. 5d, e). Since the fluorescence signals at 500 CFU mL$^{-1}$ were distinguishable from those at 0 CFU mL$^{-1}$, the limit of detection (LOD) of the Dz$^{EC}$-GMø-based assay for *E. coli* detection was estimated to be 500 CFU mL$^{-1}$ (Fig. 5e, inset), which was better than other DNAzyme-based methods (Supplementary Table 2) or the antibody-based lateral flow or dipstick devices. This enhanced sensitivity and detection efficiency were probably due to that Dz$^{EC}$-GMøs can enrich bacteria on their surface, the distance between bacteria and DNAzymes is greatly shortened. As a result, the local concentration of CEMs secreted by bacteria around the Dz$^{EC}$-GMøs is improved and the CEMs have a shorter diffusion distance to activate DNAzymes, making GMøs faster and easier to respond to analytes. The role of magnetic enrichment in bacteria detection was also verified by detecting blood samples spiked with a low-concentration *E. coli* ($10^4$ CFU mL$^{-1}$). As shown in Supplementary Fig. 19, bacteria-enriched samples showed an 8.6-fold enhancement in fluorescence, thus verifying the advantageous role of the magnetic enrichment step in bacteria analysis. In addition, a large number of impurities that exist in the biological samples (e.g., exfoliated cells, blood cells or cell debris) dramatically disrupt the analysis, while a simple magnetic enrichment and separation step can address these problems[34]. This method is highly specific, and other bacteria such as *S. aureus*, *P. aeruginosa*, *V. parahemolyticus*, *Streptococcus pyogenes* (*S. pyogenes*), and *C. albicans* could not produce detectable signals (Fig. 5f).

The sensitivity to detect a single live bacterium is a long-term pursuit for a bacterial detection method in consideration of its applications in the field of infection diagnostics and food safety. To address this problem, we developed a single Dz$^{EC}$-GMø-based assay, in which only one Dz$^{EC}$-GMø was used and analyzed after adding bacteria. In this approach, all reactions occurred on one cell particle; moreover, only a fraction of the activated DNAzymes is sufficient to produce a detectable signal, making the assay much more sensitive than traditional methods for detecting the average fluorescence intensity. To demonstrate this, CEMs secreted by *E. coli*

corresponding to 1 CFU were incubated with different concentrations of Dz$^{EC}$-GMøs (G1-G4: 1, 10, $10^2$, $10^3$ cells) for 0.5 h. After activation, cells were analyzed using a fluorescence microscope. The preparation of a single cell-based assay is shown in Supplementary Fig. 20. As shown in Fig. 5g, as the GMø number decreased, the fluorescence significantly increased and the group containing a single Dz$^{EC}$-GMø produced the highest signal, thus confirming the feasibility of our approach for single-live-cell detection.

The design principle of our GMø-based bacteria detector is highly generalizable and can be easily adapted to other bacteria by changing the used DNAzyme. To verify this, we constructed another GMø-based biosensor using a methicillin-resistant *S. aureus*-specific DNAzyme (denoted as Dz$^{SA}$-GMø)[35] and demonstrated that *S. aureus* can trigger the activation of DNAzymes on the GMø with sensitivity and specificity comparable to those observed with the Dz$^{EC}$-GMø (Supplementary Fig. 21).

Because of their specific response to corresponding targets, different DNAzymes can be decorated orthogonally on the GMø for bacteria detection without crosstalk. To verify this, we modified two DNAzymes targeting *E. coli* and *S. aureus* (termed as Dz$^{EC}_{SA}$-GMø) on the cell membrane (Fig. 5h). Each enzyme labeled with a fluorophore (FITC or rhodamine, Rh) could bind to the corresponding bacteria-secreted CEMs and cleave their substrate, resulting in the recovery of fluorescence. The introduction of one of the two bacteria in the sample lightened the specific DNAzyme and only addition of both bacteria we observed the recovery of the two fluorophores (Fig. 5i).

## Analysis of pore-forming toxins using GMøs

To further broaden the applicability of the GMø-based platform, we used it to analyze pore-forming toxins (PFTs) secreted by bacteria. α-Hemolysin (Hlα) is a well-known exotoxin that disrupts cells by forming pores in cellular membranes and causing their penetration imbalances[36]. Therefore, developing effective strategies for accurate, rapid, and sensitive detection of this toxin is of great importance for profiling the virulence of pathogens.

The sensing principle is shown in Fig. 6a. The binding event between the GMø membranes and the target toxin results in the formation of pores on the cell membrane, thus facilitating the permeation of fluorescent molecules (i.e., propidium iodide, PI), an impermeable dye to nondestructive membranes. Subsequently, the PI fluorescence in the cells could be rapidly analyzed by fluorescence microscopy or flow cytometry analysis. In the absence of Hlα, GMøs remained dark during the experiment (Fig. 6b). In contrast, red fluorescence in the GMøs was gradually observed after adding 100 nM of Hlα and reached a plateau at 25 min, suggesting the formation of pores on the membrane and permeation of PI dyes. Furthermore, when anti-Hlα antibodies were introduced into the system, the PI signal was significantly decreased (Supplementary Fig. 22), which was due to the neutralization effect of antibodies, further confirming the fluorescence of GMøs was due to the damage of Hlα toxin.

Based on the above results, we used a single GMø-based assay for the ultrasensitive detection of Hlα. Specifically, one GMø was incubated with different concentrations of toxin and then stained with PI dyes. As shown in Fig. 6c, PI fluorescence of the GMø increased as the concentration of the toxin increased. Since the fluorescence signals at 10 fM were distinguishable from the background signals, the limit of detection (LOD) of the PI staining assay for Hlα detection was estimated to be 10 fM (Fig. 6d, inset), which was significantly lower than the previously reported whole-cell-based hemolytic assay (nM range)[37], enzyme-linked immunosorbent assay (pM range)[38], fluorescence assay (pM range)[39] and field effect transistor sensors (nM)[40], further emphasizing the advantage of our approach for pathogen-associated toxin analysis (Supplementary Table 3). Of note, the intact structure of GMøs could exclude PI dyes for at least 2 weeks (Supplementary Fig. 23), which ensured its real applications. Finally, we tested

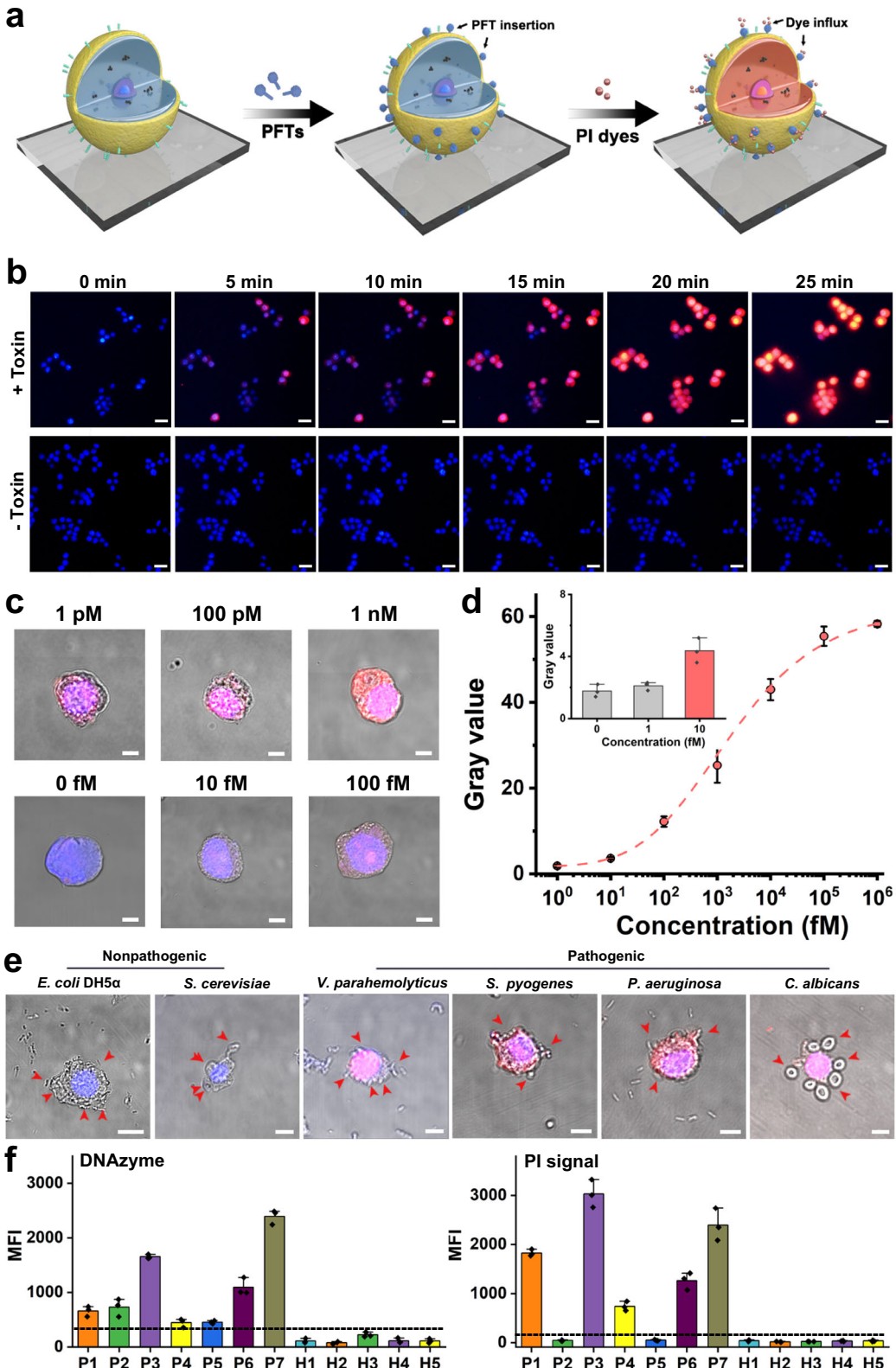

**Fig. 6 | Analysis of bacteria-secreted pore-forming toxins. a** Schematic illustration of GMø-based method for PFT analysis. **b** Real-time monitoring of the PI fluorescence of GMøs in the absence or presence of Hlα (100 nM). Scale bars: 20 μm. **c** Confocal images and **d** corresponding fluorescence intensities of a single GMø incubated with different concentrations of Hlα and subsequently stained with PI molecules. Scale bar: 5 μm. Inset shows the fluorescence intensity at low Hlα concentrations. **e** Analysis of the disruption effect of nonpathogenic or pathogenic microorganisms on the GMøs. *E. coli* DH5α and *S. cerevisiae* are mild strains that do not secrete hemolytic toxins. Red arrows show the captured microorganisms. Scale bars: 5 μm. **f** Flow cytometry analysis of *S. aureus* and PI fluorescence of sputum samples obtained from 7 pneumonia patients caused by *S. aureus* and 5 healthy individuals (control) using Dz$^{SA}$-GMø-based assays. The error bars represent mean ± SEM, $n = 3$. Source data are provided as a Source data file.

the versatility of our GMø-based assay using another bacteria-associated α-PFT, i.e., cytolysin A (ClyA). To rapidly obtain fluorescence signal, we used flow cytometry analysis to measure the PI fluorescence of GMøs after incubating with different concentrations of ClyA, and results confirmed the successful quantitatively analysis of ClyA (Supplementary Fig. 24).

The method was further employed to analyze the disruption effect of different bacteria on cell membranes. To this end, GMøs were incubated with fresh pathogenic or nonpathogenic bacteria ($10^3$ CFU mL$^{-1}$) and then subjected to a PI staining experiment. Among these microorganisms, *E. coli DH5α* and *Saccharomyces cerevisiae (S. cerevisiae)* are mild strains that do not secrete hemolytic toxins, while *V. parahemolyticus, S. pyogenes, P. aeruginosa*, and *C. albicans* are known to secrete PFTs or directly damage cell membranes. For example, *V. parahemolyticus* secretes thermostable direct hemolysin (TDH) that can form pores in the membrane lipid bilayer[41]; *S. pyogenes* produces a membrane-damaging protein, i.e., streptolysin O and *P. aeruginosa* cooperatively use exolysin (ExlA) and Type IV Pili to exert its cytotoxic activity by promoting close contact between bacteria and the host cell[42,43]. As for *C. albicans*, it forms hyphae or secretes candidalysin to cause cell membrane damage[44]. As depicted in Fig. 6e, only GMøs treated with the secretions obtained from pathogenic bacteria showed strong PI fluorescence (i.e., *V. parahemolyticus, S. pyogenes, P. aeruginosa*, and *C. albicans*), suggesting the destructiveness of these microorganisms toward immune cells. As a comparison, the hemolytic assay required at least 12 h to produce detectable signals at the same bacteria concentration (Supplementary Fig. 25).

We also used Dz$^{S4}$-GMøs to analyze real sputum samples of *S. aureus*-caused pneumonia patients collected from a local hospital. After a simple capture-and-detect step, the resulting Dz$^{S4}$-GMøs were collected by an external magnetic field and subjected to flow cytometry analysis. In this assay, the presence of *S. aureus* in the samples would activate the DNAzymes, generating green fluorescence on the cell membranes, and Hlα could punch GMøs and cause an increase in PI fluorescence. As shown in Fig. 6f, all patient samples generated a positive signal for *S. aureus* (left panel), which was ascribed to the activation of DNAzymes on the gelated cell particles. Meanwhile, PI staining experiment (right panel) showed that not all *S. aureus* secreted Hlα (i.e., P2 and P5), which was in agreement with the results of the conventional hemolytic assay (Supplementary Fig. 26). More details are shown in Supplementary Fig. 27. These results revealed that our method had the ability to rapidly analyze bacteria and toxin in clinical samples.

## Discussion

In summary, we have developed a nanoengineered approach to transform native Møs into a pathogen detector. The intracellular hydrogelation of Møs greatly improves their mechanical stability compared to native cells, and the successful magnetization of Møs by internalization of MNPs also improves their operability, thus making it possible to facilely deploy these cell particles in practical use. Because of the mild fabrication process, the resultant GMøs not only retain their recognition capacity to capture and enrich a broad-spectrum of microorganisms (e.g., bacteria and fungi) but can also be modified with DNA sensing elements for specifically reporting multiple types of bacteria in a single assay. Furthermore, taking advantage of a single-cell analysis platform, the influence of microorganisms on the immune cells can be analyzed and exotoxins secreted from pathogenic bacteria can be quantified with improved performance compared to traditional methods. Expanding the functionality of naturally existing entities is one of the exciting fields in biotechnology, our study realizes this concept without involving complex genetic operations and proposes design rules that allow many other cells and DNA elements to be jointly used as smart tools guiding cell detection and clinical diagnostics.

## Methods

### Ethical statement

All sputum samples were acquired and handled according to the protocols approved by the Scientific Ethical Committee of the First Affiliated Hospital of Nanjing Medical University (No. 2021-SPFA-360). Blood samples were collected from lab volunteers approved by the Scientific Ethical Committee of Tongren Hospital of Shanghai (No. 2023-014-01). Informed consent was obtained and no compensation was provided for all research participants.

### Materials

All used oligonucleotides (Supplementary Table 1), phosphate-buffered saline (PBS), Dulbecco's modified Eagle medium (DMEM), LB Broth medium, and phenol-red-free DMEM were purchased from Sangon Biotechnology Co., Ltd. (Shanghai, China). RAW264.7 and MCF-7 cells were purchased from American Type Culture Collection (ATCC, Manassas, VA, USA). CD206 Polyclonal antibody (catalog number: 18704-1-AP), CD163 polyclonal antibody (catalog number: 16646-1-AP), F4/80 polyclonal antibody (catalog number: 29414-1-AP), and rabbit IgG control polyclonal antibody (catalog number: 30000-0-AP) were obtained from Proteintech Group (Rosemont, USA). TLR2 polyclonal antibody (catalog number: orb191498), TLR4 polyclonal antibody (catalog number: orb371961) were ordered from Biorbyt (Cambridge, UK). Anti-*E. coli* monoclonal antibody (catalog number: sc-57709, clone number: 1011), anti-MRSA monoclonal antibody (catalog number: sc-73327, clone number: NYR MRSA16), and Alexa Fluor® 594-labeled CD206 monoclonal antibody (catalog number: sc-58986, clone number: 15−2) were ordered from Santa Cruz Biotechnology (Dallas, USA). Alpha Hemolysin (hly) antibody (catalog number: abx109435) was purchased from Abbexa (Cambridge, UK). Magnetic nanoparticles (diameter = 200 nm) were purchased from BaseLine Chromtech Research Centre (Tianjin, China). Silica microbeads (SiMBs, diameter = 10 μm), diamidinyl phenyl indole (DAPI), prodium iodide (PI), 2-(N-morpholino)ethanesulfonic acid (MES), N-(3-Dimethylaminopropyl)-N′-ethylcarbodiimide hydrochloride crystalline (EDC), N-Hydroxysuccinimide (NHS), and bovine serum albumin (BSA) were supplied by Aladdin (Shanghai, China). All strains were purchased from China General Microbiological Culture Collection Centre. Interleukin-4 (IL-4) was ordered from GenScript Biotech Corporation (Nanjing, China). 2-hydroxy-4′-(2-hydroxyethoxy)-2-methylpropiophenone, poly(ethylene glycol) diacrylate (Mw = 700 Da, PEG700-DA), α-hemolysin, and fetal bovine serum were obtained from Sigma−Aldrich (Shanghai, China). Cytolysin A was obtained from NovoPro (Shanghai, China). Fc fragment was purchased from AmyJet Scientific (Wuhan, China). Mouse FcR Blocking Reagent was ordered from NovoBio-technology Co., Ltd. (Beijing, China). CCK-8 kit and prodium Iodide (PI) was purchased from Beyotime Biotechnology (Shanghai, China). Sputum sample dilution was purchased from Panglong Medical devices (Chongqing, China). Deionized and RNase-free water (resistance >18 MΩ·cm) was used throughout the experiments.

### Cell culture

RAW264.7 and MCF-7 cells were cultured in culture dish (diameter = 10 cm) and placed in a cell incubator with 37 °C, 5% CO$_2$. DMEM supplemented with 10% fetal bovine serum was used as culture medium. For magnetization and activation, RAW264.7 cells ($5 \times 10^6$) were simultaneously treated with 20 ng mL$^{-1}$ of IL-4 and 80 μg mL$^{-1}$ of MNPs for 12 h at 37 °C and then washed with PBS for subsequent use.

### Flow cytometry analysis

For analyzing the surface expression level of protein markers (i.e., TLR2, TLR4, CD206, and CD163) on IL-4 treated RAW264.7 macrophage cells, the cells were cultured in 6-well plate for 24 h and incubated with IL-4 (20 ng mL$^{-1}$) overnight at 37 °C. Then the cells were collected and stained with corresponding antibodies (i.e., Rabbit IgG

control, anti-TLR2, anti-TLR4, anti-CD206, and anti-CD163 antibodies, 15 μg mL$^{-1}$) diluted in PBS for 20 min on ice. After that, the cells were washed with ice PBS and centrifuged at 660 $g$ for 3 min, then the surface expression of protein receptors was calculated by a flow cytometer (BD FACS Verse, USA) and the data was analyzed using Flowjo V10.0 software. The number of events analyzed was 10,000 per sample. The exemplification of gating strategy was shown in Fig. S28.

## Inductively coupled plasma mass spectrometry (ICP-MS)
The cells were seeded in 12-well plates and then incubated at 37 °C for 48 h until they reached 70–80% confluence. The supernatant was then removed and replaced with 1 mL of MNP solution for 2, 4, 8, 12, and 24 h. After removing excess nanoparticles, the cells were washed three times with PBS, harvested, and counted using the hemacytometer. The resulting cell solutions were diluted to a total volume of 10 mL with ultra-clean 2% nitric acid and sonicated in an ultrasound bath for 30 min at 45 °C. The digested iron contents were measured by Varian 820-MS mass spectrometer and the average MNP numbers internalized by one cell were calculated.

## Bacteria strains
Methicillin-resistant *Staphylococcus aureus* (MRSA, ATCC33591), *Listeria monocytogenes* (ATCC15313), *Escherichia coli* K12 (*E. coli*, ATCC700728), *Escherichia coli* DH5α (*E. coli* DH5α), *Pseudomonas aeruginosa* (*P. aeruginosa*, ATCC15313), *Salmonella enteritidis* (*S. enteritidis*, ATCC19585), *Streptococcus pyogenes* (*P. pyogenes*, ATCC12344), *Candida albicans* (*C. albicans*, ATCC10231) were cultured in sterile LB broth for 12 h at 37 °C. After that, bacteria were centrifuged and collected at 2600 × $g$ for 5 min.

## Intracellular hydrogelation of RAW264.7 cells
The hydrogelation of macrophages was according to previous reports. Briefly, 1 mL of gelation buffer was prepared by mixing 100 μL of 2-hydroxy-4'-(2-hydroxyethoxy)-2-methylpropiophenone (1.5 g mL$^{-1}$) and 900 μL of PEG700-DA. In parallel, 5 × 10$^6$ Møs were collected and suspended in 500 μL of phenol-red-free DMEM containing 1× protease inhibitor. Then, 1 mL of the gelation buffer was added to the cell suspension to reach a 10 wt% PEG700-DA concentration. For fluorescent labeling of the gelated cells, fluorescein O, O'-diacrylate was added into the gelation buffer. After 5 min incubation, the cells were centrifuged and resuspended in 500 μL phenol-red-free DMEM without gelation buffer and subjected to 365 nm irradiation for 5 min using a UV oven (UVP Crosslinker, CL3000, USA). Of note, a 365 nm UV lamp was used to excite the photoinitiator as the wavelength reduces protein denaturation and cell toxicity. The resulting gelated cells were collected and washed with PBS for subsequent analysis.

## DNA decoration
Equal amounts of DNA1/DNA2 or DNAzyme/substrate (final concentration: 100 nM) were added to the PBS solution, heated at 90 °C for 1 min, then cooled to room temperature for 10 min, followed by the addition of GMøs (1 × 10$^7$ cells). After 15 min incubation, excess DNA was removed by magnetic separation and the resulting DNA-GMøs were stored at 4 °C for subsequent use. To determine the DNA number on each cell, the fluorescence spectra of DNA1/DNA2 (100 nM) before and after incubating GMøs (1 × 10$^7$ cells) were recorded using an F-7100 fluorophotometer (Hitachi, Japan) with an excitation wavelength of 488 nm and emission wavelength of 520 nm and then calculated according to the standard curve of FITC-DNA.

## TEM analysis
Cellular samples were fixed using 2% glutaraldehyde solution overnight at 4 °C. After post-fixation in 1% osmium tetroxide and pre-embedding staining with 1% uranyl acetate, tissue samples were dehydrated and embedded in Agar 100. Sections measuring 80 nm were then examined using a JEM-1400Flash microscope (JEOL, Japan).

## Stability investigation
Different cells (1 × 10$^6$ cells) were incubated in the 1× PBS for 30 min, NaCl solution (1 M) for 15 min, and pure water for 30 min with occasional shaking. Also, the same concentration of cells was treated in a KQ-50E ultrasonic machine (40 W, Kunshan, China) for 10 min, stored for 30 days at room temperature, and centrifugation (10,600 × $g$) for 10 min and redispersion four times. For freeze–thaw cycles, cells were subjected to a standard programmed cryopreservation process repeated three times. After that, frozen cells were thawed in a 37 °C water bath for microscopy analysis, and the residual cell number was counted using a hemocytometer. For each sample, cells stored in the PBS was used as the positive control. Cell retention (%) = Cell number$_{treatment}$/Cell number$_{PBS}$.

After modification of fluorescent DNA strands, the resulting DNA-GMøs (2 × 10$^5$ particles) were incubated with 10% bovine serum for different times (0, 1, 2, and 4 h). At each time point, cell particles were analyzed by a flow cytometer (BD FACS Verse, USA). To investigate the stability of protein markers, GMøs (2 × 10$^5$ particles) after long-term storage were incubated with anti-CD206/anti-CD163 antibodies (15 μg mL$^{-1}$) for 0.5 h at room temperature and then subjected to flow cytometry analysis. The number of events analyzed was 10,000 per sample.

## Fluidity and integrity investigation of the GMøs
Gelated cell samples were stained by adding DiI dye solution (final concentration: 4 μM) and seeded on a polylysine-coated glass slide to avoid random movements. Then, fluorescence recovery after photobleaching (FRAP) experiment was performed to assess the membrane fluidity using a Zeiss LSM880 confocal microscope (Germany). The sample was first measured thrice (5% laser power) to take the fluorescence pictures before photobleaching, followed by laser pulses with full power to bleach a selected area (green rectangular box) at the plasma membrane. Fluorescence recovery was recorded every 1.26 s until a plateau was reached (ca 120 s). Fluorescence intensity vs. time was plotted for analyzing the fluorescence recovery. Confocal images were analyzed using ZEN V2.3 SP1 (blue edition) software.

For investigating the membrane integrity, 40 μL of the freshly prepared GMøs (1 × 10$^3$ cells) and 10 μL of DAPI (final concentration: 5 μM) and PI solution (final concentration: 2 μM) were mixed and incubated for 20 min. Then, the cell particles were separated using an external magnet and dropped on a glass slide for fluorescence microscopy analysis (Carl Zeiss Axio A1, Germany).

## Cell viability assay
The cytotoxicity of MNPs was evaluated using the CCK-8 kit. Raw 264.7 cells were cultured on a 96-well plate (1 × 10$^4$ cells/well) with 100 μL complete culture medium in a humidified atmosphere incubator of 5% CO$_2$ at 37 °C. After 12 h, MNPs (final concentration: 80 μg mL$^{-1}$) were added into the 96-well plate and cultured with cells for another 24 h. After washing, cells were treated with the 10% (v/v) CCK-8 reagent in DMEM medium for 1 h at 37 °C, and the absorbance at 450 nm was recorded by a microplate reader (TECAN Infinite M200, Switzerland). The cells treated with PBS were set as control groups. To evaluate the viability of GMøs, GMøs were immediately subjected to CCK-8 analysis after gelation.

## Bacteria capture
In all, 50 μL of GMø (1 × 10$^5$ cells) solution was incubated with 50 μL of bacteria suspension (final OD$_{600\,nm}$ = 0.2–0.3) for 15 min at room temperature with continuous shaking (350 rpm). Then, the cell particles were separated using a magnet and washed with PBS twice. After that, 5 μL of GMø solution was subjected to confocal microscopy analysis (Carl Zeiss LSM880, Germany) or a standard spread plate count assay after mild sonication and dilution, followed by colony count after 48 h using an image J 1.52 software.

To perform the antibody-blocking assay, GMøs ($1 \times 10^5$ cells) were treated with FcR Blocking Reagent for 40 min. Then, the potential bacteria-binding receptors were blocked by corresponding antibodies (i.e., anti-CD206 antibody, 20 µg mL$^{-1}$, anti-CD163, 15 µg mL$^{-1}$, and anti-F4/80 antibodies, 12 µg mL$^{-1}$) for 1 h with continuous shaking (350 rpm). In parallel, *S. aureus* was incubated with mouse Fc fragments (2 mg mL$^{-1}$) to block protein A on the bacterial surface. After that, the blocked GMøs and *S. aureus* were used for bacteria capture and the captured number of bacteria was counted by a plate assay.

To prepare antibody-modified SiMBs (Ab-SiMBs), 1 mL of the carboxylated SiMBs ($1 \times 10^7$ particles, $d = 10$ µm) were diluted in MES (0.05 M, pH 6.0). 1 mg EDC and 1.5 mg NHS were added to the solution and allowed to react for 30 min with gentle mixing to activate the carboxyl groups on the SiMBs. After that, the microbeads were washed twice to remove excess reagents and dispersed in 0.5 mL of 1× PBS. 0.5 mL of anti-*E. coli*/anti-MRSA antibody solution (1 mg mL$^{-1}$) was added into the above solution and shaken for 2.5 h at room temperature. The conjugates were centrifuged and suspended in 1 mL of 3% BSA (w/v) to block the unoccupied sites for 1 h at room temperature. The saturated anti-*E. coli*/anti-MRSA antibody number on each bead was estimated to be $5.8 \times 10^5$ antibodies/bead, which was comparable to the number of protein receptors on the cells. The final products were centrifuged and stored in PBS at 4 °C before use. To capture bacteria, 50 µL of Ab-SiMBs ($1 \times 10^5$ particles) solution was incubated with 50 µL of bacteria suspension (final OD$_{600\ nm}$ = 0.2–0.3) for 15 min at room temperature with continuous shaking (350 rpm). Then, the Ab-SiMBs were separated by an external magnet and washed with PBS twice. After that, 5 µL of Ab-SiMB solution was subjected to confocal microscopy analysis (Carl Zeiss LSM880, Germany).

For collecting bacteria from complex media, GMøs or Ab-SiMBs were used to capture *E. coli* and *S. aureus* spiked in 10% human plasma, 10% human serum, 20% saliva, and 20% urine in the same way as described above.

### Bacteria detection

For flow cytometry analysis, 90 µL of DNAzyme modified-GMøs (Dz-GMøs, $1 \times 10^4$ cells) in a reaction buffer (50 mM HEPES, 150 mM NaCl, 15 mM MgCl$_2$, pH 7.4) was mixed with 10 µL of *E. coli* K12 or MRSA with varying concentrations. After 30 min of reaction, cell particles were collected and subjected to confocal microscopy and flow cytometry analysis (BD FACS Verse, USA). To confirm the specificity of the method, 10 µL of different microbes (final concentration: $1 \times 10^7$ CFU mL$^{-1}$) was mixed with 90 µL of Dz-GMøs ($1 \times 10^4$ cells) for 30 min. Then, the fluorescence signal was detected using a flow cytometer.

For single-cell analysis, one GMø was collected and transferred into 4 µL of reaction solution on a glass slide using a Narishige micromanipulator system (Supplementary Fig. 20). Then, 1 µL of *E. coli*-released CEMs corresponding to one CFU was prepared and added to the reaction buffer. After 30 min reaction, cell particles were collected and subjected to fluorescence microscopy analysis (Carl Zeiss Axio A1, Germany).

### PI experiments

For real-time analysis of Hlα, 100 µL of GMøs ($1 \times 10^4$ cells) was added in a 96-well plate and then incubated with PBS solution containing Hlα (100 nM) and PI molecules (4 µM) for 25 min. The image at the DAPI channel of the samples was recorded at each time point (0, 5, 10, 20, and 25 min) using a fluorescence microscope.

For the single cell particle-based assay, one GMø was prepared using the same procedures described above. Then, 1 µL of samples containing different concentrations of Hlα and PI molecules (4 µM) was added and the pictures were collected using a fluorescence microscope after 25 min incubation.

To confirm the role of Hlα, 5 µL of Hlα solution (200 nM) was treated with blocking antibodies (anti-SR or anti-Hlα antibodies,

20 µg mL$^{-1}$) for 30 min. After that, 95 µL of GMøs ($1 \times 10^4$ cells) was mixed with the antibody-treated Hlα and PI (4 µM) solution for 25 min. Then, the solution was subjected to flow cytometry analysis.

To study the disruption effect of microorganisms on the GMøs, 9 µL of cell particles ($1 \times 10^2$ cells) were first incubated with 1 µL of microbes (final concentration: $10^4$ CFU mL$^{-1}$) for 15 min with mild pipetting. After dropping the reaction solution on the glass slide, PI molecules (4 µM) were added and the images were collected using a fluorescence microscope after 15 min staining.

### Hemolytic assay

Fresh blood samples were collected from lab volunteers and the red blood cells were collected by low-speed centrifugation. Then, 650 µL of blood cells were incubated with 50 µL of different microbes (final concentration: $10^3$ CFU mL$^{-1}$) for at least 12 h. To investigate the pore-forming toxins (PTFs) secreted by bacteria in the sputum samples, the samples were washed, homogenized, and cultured overnight. After that, the culture supernatants were collected and used for the hemolytic assay.

### Real samples analysis

In all, 100 µL of 50%, 25%, and 12.5% sputum was prepared by diluting the samples with sputum sample dilution. Since the viscosity of the 25% sputum sample was suitable for bacteria capture and signal processing, this concentration was chosen for all the spiking experiments. 10 µL of Dz$^{SA}$-GMøs ($2 \times 10^4$ cells) were incubated with 10 µL of diluted sputum samples containing PI molecules (4 µM) for 60 min with continuous shaking. After collecting the cell particles by an external magnet, the activated DNAzymes (FITC channel) and PI fluorescence of the GMøs were measured by a standard flow cytometry analysis. To test the secretion of PTFs by bacteria in the sputum samples, 10 µL of diluted sputum samples were incubated with 90 µL of solution containing blood cells and 1% serum for 12 h.

### Statistics and reproducibility

Statistical analysis and plotting of data were performed using OriginPro 2019. One-way analysis of variance (ANOVA) for multiple comparisons were used in Fig. 2b. The DNA stability was compared using paired-sample *t* test analysis in Fig. 2c. In Fig. 4c, e, bacteria colony was compared using two-sample *t* test analysis. Data represent mean ± SEM; n is stated in the figure legend. In Fig. 2c–f; 4a; 5b, g, h; and 6b–e and Supplementary Figs. 1b; 8a, b; 10; 11; 16; 21a; and 24a, b, all data was obtained from at least three independent experiments. OriginPro 2019 was used for statistical analysis, and cell images were analyzed by Image J v1.52. Confocal images were analyzed by ZEN v2.3 SP1 (blue edition). Flow cytometry data was analyzed by FACSDiva v9.0 software and Flowjo v10.0.

### Reporting summary

Further information on research design is available in the Nature Portfolio Reporting Summary linked to this article.

## Data availability

All data generated in this study are provided in the Supplementary Information and Source data files. Data can also be provided by the corresponding author upon request. Source data are provided with this paper.

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

## Acknowledgements

This work was supported by the National Natural Science Foundation of China 21804069 (C.L.) and the Fundamental Research Funds for the Central Universities of China JZ2021HGTB0120 (C.L.).

## Author contributions

C.L. conceived and designed the experiments. Y.G. and Y.Z. performed the experiments. Y.G., Y.Z., B.C., Y.Y., and J.M. analyzed the data. C.L. and Y.Z. wrote the manuscript. All authors reviewed the manuscript and approved the manuscript.

## Competing interests

The authors declare no competing interests.
