## [Peer Review File · Nature Communications]

REVIEWER COMMENTS

Reviewer #1 (Remarks to the Author):

In this article “A Smart Pathogen Detector Engineered from Intracellular Hydrogelation of DNA-Decorated Macrophages”, the authors leverage a recently developed cell gelation technique to develop gelled macrophages (GMØs) for pathogen recognition. Three distinctive modalities are proposed: 1.) magnetic GMØs for bacteria enrichment. 2) DNAzyme-conjugated GMØs for strain-specific bacteria detection. 3) A PI-based fluorescence approach for pore-forming toxin detection. Notably, the authors tested the PFT sensing modality of the GMØ with sputum samples collected from *S. aureus*-caused pneumonia patients. The article is interesting in leveraging an emerging biotechnology for novel pathogen-sensing design. However, due to the three distinctive modalities proposed by the authors, the article is overextended and lacks sufficient depth and backgrounds for each of the proposed modality. In addition, characterizations of the GMØs, particularly elements related to bacterial engagement can be bolstered. Specific comments are as follows:

Major points

General info:

1. Due to the three distinctive modalities proposed by the authors, the background does not sufficiently describe the challenges each modality is aimed at tackling. To some extent the three distinctive modalities are antagonistic to each other. For instance the latter two modalities have no need for magnetic extraction. How each modality would offer its diagnostic value and potentially complement each other should be described.
2. The introduction should include descriptions regarding features and backgrounds of the gelled cell technology in recent references [Nat. Commun. 10, 1057, 2019][Adv. Mater. 33, 2101190, 2021]. In addition, as these prior work provide many of the evaluation techniques adopted in the present study (i.e. water test), proper citation should be added. Proper referencing of these prior work is critical to establish the foundation for the present study, which concerns primarily about new applications of the gelled cell system and has relatively shallow characterizations and discussion of the gelled cells' inherent properties.
3. The specific choice of RAW264.7 should be described as opposed to alternative cellular options. Why choose a cell line of murine origin rather than that of human origin. One would expect that in clinical testing RAW264 cells are more susceptible to complement activation and membrane perforation in human serum.

4. Mannose receptor is the sole membrane protein the authors highlight as a bacteria ligand. However, there are many other receptors and pattern recognition receptors (PRRs), such as Toll-like receptors, scavenger receptors, glycan receptors, on the surface of M ϕ that take part in bacterial recognition (Ref: Annu. Rev. Immunol. 23:901–44, 2005). In the context of the present work, these ligands relevant to bacterial binding should be characterized on the GM ϕ s. The present manuscript in fact has no membrane protein characterization of the GM ϕ s and only shows mannose receptors on M ϕ s before gelation in Figure 1B.

5. The introduction of DNA duplex in Figure 2 and the adoption of DNA-GM ϕ s for bacteria capture in Figure 3 are very confusing. The way the DNA is introduced makes it appear as a key component for bacteria binding. Upon close scrutiny, however, the DNA only serves in the DNAzyme detection modality, which makes its appearance in Figure 3 regarding bacteria capture unnecessary and distracting. The data reorganized for better clarity.

Bacteria Capture modality:

6. The capture modality in Figure 3 only shows qualitatively the differences between GM ϕ s with and without IL-4 stimulation. Yet there is no quantitative evaluation regarding the GM ϕ s' capture efficiency. For instance, a titration study with varying GM ϕ s is important in assessing the affinity of the system and provide a picture on the number of GM ϕ s one intends to use in a given volume of clinical sample (i.e. blood or urine). In addition, improvement of detection limit before and after designated GM ϕ s-mediated enrichment should be demonstrated. The authors can refer to prior references for experimental designs aimed at demonstrating pathogen enrichment for enhanced detection (ACS Appl. Mater. Interfaces 2017, 9, 39953–39961).

7. Comparison between GM ϕ s and Ab-conjugated magnetic beads in figure 3E needs proper standardization. Gelated cells and magnetic beads have significantly different surface area, density, and surface ligand number. Proper standardization with sound justification are needed for relevant comparison.

DNAzyme detection modality

8. In Figure 4, the combination of GM ϕ inserted with specific-designed DNAzyme system for *E. coli* or *S. aureus* is promising with good sensitivity. However, as the design is aimed at improving a previously developed fluorogenic diagnosis technology, (Ref: J Vis Exp. (63): 3961., 2012), sensitivity improvement by the GM ϕ system over free DNAzyme should be demonstrated. It is presently unclear how much improvement is afforded by the GM ϕ system over the free DNAzyme approach.

9. The robustness of the DNAzyme detection modality should be demonstrated in physiologically relevant media in order to show the practical value of the device.

PI-based PFT and pathogenic bacteria detection assay modality

10. The adoption of the GM ϕ for detecting membrane-damaging virulence factors is interesting, but the ambitious claims laid out by the authors require additional supporting data. Firstly, the proposed detection assays have existing alternatives using hemolytic assays for PFT and hemagglutination assays for membrane-binding bacteria. The sensitivity of the proposed system over these common approaches should be compared.

11. A singular β -PFT is used in the present study, yet there are two major classes of PFTs, α -PFTs and β -PFTs, and different class families recognize their target host cells with high specificity (Ref: Nat Rev Microbiol. Feb;14(2):77-92, 2016). These PFTs recognize specific lipid constituents, sugar moieties, or protein receptors of the target membrane. Thus more types of PFTs should be included to define the applicability of the GM ϕ .

12. The depiction regarding Figure 5E is unclear. It is unclear whether the authors propose that the PI-based method can detect pathogenic bacteria because they can secrete PFTs or because they can damage cell membrane upon adherence. Such clarification is critical as it concerns whether the presence of bacteria is needed during sample analysis.

13. Quantitative analysis for pathogenic bacteria detection should be performed.

14. If detection of pathogenic pathogen is based on PFT, specific PFTs for the pathogenic bacteria should be described and evaluated in parallel.

15. The context of the GM ϕ -based diagnosis with patient sputum needs to be better defined with alternative assay validation. For instance, sputum sample should also be analyzed using hemolysis assay. For the current data in Figure 5F, it is unclear if the GM ϕ has a detection accuracy of 5 out of 7 (~70%) or if GM ϕ is able to distinguish specific features associated with different clinical samples.

16. Are the GM ϕ able to exclude PI indefinitely? Or is there a time window and preservation protocol for preserving GM ϕ membrane integrity? Such information needs to be provided to convey clinical translationability of the system.

Minor points

17. For all data in this manuscript, authors need to describe their statistical method and whether all data are representative of at least three independent experiments in some figure legends (Fig. 3E, Fig. 4I). Before choosing method, they should test whether their data is normally distributed before deciding what statistical test to carry out.

18. IRB protocol number is missing for the use of human sputum.

19. Bacteria number in Figure 5F is strange. How are the authors able to detect low bacteria number in the single digit in clinical samples. It is likely a mislabel.

Reviewer #2 (Remarks to the Author):

This work describes a novel experimental approach to improve pathogen detection by exploiting the ability of macrophages to capture pathogens. Macrophages (cell line RAW264.7 cells) are made resistant to harsh handling conditions through intracellular hydrogelation, and are loaded with pathogen-specific functional nucleic acids. Further, internalisation of magnetic nanoparticles by macrophages facilitates enrichment using a magnet.

These cells are used to detect pathogens and toxins. Toxins are detected using propidium iodine (PI) that only penetrates damaged cells.

I won't comment on the efficiency of pathogen detection compared to other methodologies or the methodology used to incorporate hydrogels into the cells as they are out of my area of expertise.

General comments:

The authors focus on MR as an important pathogen-capture mechanism and exploit IL-4 to increase MR expression (Figure 1), which correlates with increased pathogen capture by IL-4-treated cells (Figure 2). Authors should consider that CD206 does not bind all pathogens and more extensive analysis of surface markers and how they are affected by IL-4 and the gelation process is required. Particularly because of the unusual pattern of bacteria binding shown in Figure 3 A (how representative are these images?). It is unclear if any bacteria has been internalised and there is no information on the mechanism of binding. Membrane (and MR) labelling should be used to delineate cellular membrane and endosomal compartment. Is bacteria binding inhibited by specific sugars? Does it require Calcium? Previous work with dendritic cells indicate that membrane features are maintained after the gelation process (references 24 and 25) and this was also confirmed by this work but as far as I know no endocytosis or phagocytosis after gelation has been demonstrated. Is it worth looking at this?

Technical/presentation queries:

What would happen if a non-macrophage cell line is used? Apart from lack of uptake of magnetic particles, which presumably requires phagocytosis, would bacteria attach to these cells after gelation?

There is no quantification of the efficiency of macrophage recovery after magnetic purification.

PI is routinely employed in flow cytometry. I would suggest testing toxicity using this technique, which will facilitate quantification.

Use of pathogen mutants lacking expression of toxins would strengthen the toxicity data.

Is candidalysin responsible of *C. albicans*-cytotoxicity? Could authors comment?

Figure 1F should include TEM of untreated macrophages.

The authors show the stability of the cells but, as far as I can tell, do not show if they maintain their pathogen binding activity. This would be essential if this technique is to be implemented in a clinical lab. Are expression of CD206 and binding activity of the cells reduced upon storage? Would they need to add additional IL-4?

IL-4 has been shown to regulate numerous markers in macrophages and act as cellular survival factor. Are its effect just due to the fact that cells are just healthier?

It would be worth testing a series of cellular markers in the presence and absence of gelation in untreated and IL-4-treated cells.

Blocking Ab experiment in Figure S6 needs further explanation. There is no control antibody and, to my knowledge, neither *E. coli* or *S. aureus* bind MR. Further, *S. aureus* could bind antibodies through Fc portion due to protein A. Anti-MR antibodies used, do not seem validated for blocking its activity.

The experiments using antibody-coated particles is poorly described.

It is unclear how data in Figure 5F were generated. Is there any normalisation? Number of bacteria seem very low. According to supplementary information "After collecting GMØs using an external magnet, the adsorbed number and PI fluorescence of bacteria were measured with the help of a confocal microscope". Was this done by CFU or using *S. aureus*-specific DNAzyme?

Number of repeats in supplementary information and some figures in main document are missing.

Reviewer #3 (Remarks to the Author):

In this manuscript, the authors present a modified macrophage-based bacterial capture platform that incorporates DNAzyme decoration to allow for the detection of target bacterial strains. The basis of the proposed platform involves modifying macrophages via hydrogelation to create gelled macrophages (GMØ) with improved strength and stability. Magnetic nanoparticles were added to the GMØs to permit

collection. The characterization and bacterial capture capabilities of the platform was evaluated via confocal microscopy with fluorescent staining, TEM, bacterial cultures, and flow cytometry. The developed GMØs exhibited no change in viability following the internalization of magnetic particles, and showed improved viability when exposed to a range of harsh environmental conditions. Bacterial assays indicated successful bacterial capture and specific detection following the incorporation of DNAzymes. Lastly, the authors also assessed pore-forming toxin detection from bacterial secretions through the dye-based monitoring of GMØ membrane destruction. I think the idea is very interesting and novel, however there major concerns to be addressed before the manuscript can be considered for publication in Nat. comm.:

- The concentration of internalized magnetic particles should be evaluated especially since the colour change is being used to confirm successful magnetization of the macrophages. Specifically, the optical properties of the solutions shown in Figure 1a should be quantified.
- Figure 2b should be reorganized to better convey trends between the three different macrophages. Statistical significance should also be included on the graph.
- Since photoinitiator is used to facilitate the crosslinking, addition of the fluorescent monomer without the photoinitiator should be assessed to confirm successful crosslinking.
- The intensities shown in Figure 2c should be quantified.
- What do the authors mean by differences in topography associated with increased bacterial aggregations around the cells? Is this indicative of increased MR presence or other changes in cell topography? If other changes are present, topographical characterization should be considered.
- Figure 3c and 3e should use log scale to better represent the differences between tested conditions.
- The data presented in Figures 3c show that the included control samples successfully capture a relatively high concentration of bacteria. How much of this bacterial capture is from insufficient washing, yielding transfer of uncaptured bacteria into the quantified test solution? If non-specific transfer is occurring, then these numbers are inaccurate. The authors should evaluate if the cell numbers presented in 3c decrease significantly with increased washing.
- Under a similar premise, the data presented in S6 should include unstimulated MØs as a control given that antibody-mediated blocking will be more effective against the innate level of MRs.
- The limit of detection should be experimentally derived.
- Moreover, the authors claim that their LOD is better than other previous works using the same DNAzyme. This claim should be better substantiated with mention of previously reported LODs.
- Given that the toxins discussed in Figure 5 are present within MRSA, why did the lysis of the GMØs not influence the DNAzyme sensing results?
- Schematic Figure 5a should be more detailed to provide better context to the modifications performed.
- Confocal images obtained for 5f should be included.

- Tests with the patient samples should also evaluate DNAzyme signals and demonstrate a practical application.

REVIEWER COMMENTS

Reviewer #1 (Remarks to the Author):

In this article “A Smart Pathogen Detector Engineered from Intracellular Hydrogelation of DNA-Decorated Macrophages”, the authors leverage a recently developed cell gelation technique to develop gelated macrophages (GMØs) for pathogen recognition. Three distinctive modalities are proposed: 1.) magnetic GMØs for bacteria enrichment. 2) DNAzyme-conjugated GMØs for strain-specific bacteria detection. 3) A PI-based fluorescence approach for pore-forming toxin detection. Notably, the authors tested the PFT sensing modality of the GMØ with sputum samples collected from *S. aureus*-caused pneumonia patients. The article is interesting in leveraging an emerging biotechnology for novel pathogen-sensing design. However, due to the three distinctive modalities proposed by the authors, the article is overextended and lacks sufficient depth and backgrounds for each of the proposed modality. In addition, characterizations of the GMØs, particularly elements related to bacterial engagement can be bolstered. Specific comments are as follows:

Major points

General info:

1. Due to the three distinctive modalities proposed by the authors, the background does not sufficiently describe the challenges each modality is aimed at tackling. To some extent the three distinctive modalities are antagonistic to each other. For instance the latter two modalities have no need for magnetic extraction. How each modality would offer its diagnostic value and potentially complement each other should be described.

2. The introduction should include descriptions regarding features and backgrounds of the gelated cell technology in recent references [Nat. Commun. 10, 1057, 2019][Adv. Mater. 33, 2101190, 2021]. In addition, as these prior work provide many of the evaluation techniques adopted in the present study (i.e. water test), proper citation should be added. Proper referencing of these prior work is critical to establish the foundation for the present study, which concerns primarily about new applications of the gelated cell system and has relatively shallow characterizations and discussion of the gelated cells' inherent properties.

3. The specific choice of RAW264.7 should be described as opposed to alternative cellular options. Why choose a cell line of murine origin rather than that of human origin. One would expect that in clinical testing RAW264 cells are more susceptible to complement activation and membrane perforation in human serum.

4. Mannose receptor is the sole membrane protein the authors highlight as a bacteria ligand. However, there are many other receptors and pattern recognition receptors

(PRRs), such as Toll-like receptors, scavenger receptors, glycan receptors, on the surface of MØ that take part in bacterial recognition (Ref: Annu. Rev. Immunol. 23:901–44, 2005). In the context of the present work, these ligands relevant to bacterial binding should be characterized on the GMØs. The present manuscript in fact has no membrane protein characterization of the GMØs and only shows mannose receptors on MØs before gelation in Figure 1B.

5. The introduction of DNA duplex in Figure 2 and the adoption of DNA-GMØs for bacteria capture in Figure 3 are very confusing. The way the DNA is introduced makes it appear as a key component for bacteria binding. Upon close scrutiny, however, the DNA only serves in the DNAzyme detection modality, which makes its appearance in Figure 3 regarding bacteria capture unnecessary and distracting. The data reorganized for better clarity.

Bacteria Capture modality:

6. The capture modality in Figure 3 only shows qualitatively the differences between GMØs with and without IL-4 stimulation. Yet there is no quantitative evaluation regarding the GMØs' capture efficiency. For instance, a titration study with varying GMØs is important in assessing the affinity of the system and provide a picture on the number of GMØs one intends to use in a given volume of clinical sample (i.e. blood or urine). In addition, improvement of detection limit before and after designated GMØs-mediated enrichment should be demonstrated. The authors can refer to prior references for experimental designs aimed at demonstrating pathogen enrichment for enhanced detection (ACS Appl. Mater. Interfaces 2017, 9, 39953–39961).

7. Comparison between GMØs and Ab-conjugated magnetic beads in figure 3E needs proper standardization. Gelated cells and magnetic beads have significantly different surface area, density, and surface ligand number. Proper standardization with sound justification are needed for relevant comparison.

DNAzyme detection modality

8. In Figure 4, the combination of GMØ inserted with specific-designed DNAzyme system for *E. coli* or *S. aureus* is promising with good sensitivity. However, as the design is aimed at improving a previously developed fluorogenic diagnosis technology, (Ref: J Vis Exp. (63): 3961., 2012), sensitivity improvement by the GMØ system over free DNAzyme should be demonstrated. It is presently unclear how much improvement is afforded by the GMØ system over the free DNAzyme approach.

9. The robustness of the DNAzyme detection modality should be demonstrated in physiologically relevant media in order to show the practical value of the device.

PI-based PFT and pathogenic bacteria detection assay modality

10. The adoption of the GMØ for detecting membrane-damaging virulence factors is

interesting, but the ambitious claims laid out by the authors require additional supporting data. Firstly, the proposed detection assays have existing alternatives using hemolytic assays for PFT and hemagglutination assays for membrane-binding bacteria. The sensitivity of the proposed system over these common approaches should be compared.

11. A singular β -PFT is used in the present study, yet there are two major classes of PFTs, α -PFTs and β -PFTs, and different class families recognize their target host cells with high specificity (Ref: Nat Rev Microbiol. Feb;14(2):77-92, 2016). These PFTs recognize specific lipid constituents, sugar moieties, or protein receptors of the target membrane. Thus more types of PFTs should be included to define the applicability of the GM \emptyset .

12. The depiction regarding Figure 5E is unclear. It is unclear whether the authors propose that the PI-based method can detect pathogenic bacteria because they can secrete PFTs or because they can damage cell membrane upon adherence. Such clarification is critical as it concerns whether the presence of bacteria is needed during sample analysis.

13. Quantitative analysis for pathogenic bacteria detection should be performed.

14. If detection of pathogenic pathogen is based on PFT, specific PFTs for the pathogenic bacteria should be described and evaluated in parallel.

15. The context of the GM \emptyset -based diagnosis with patient sputum needs to be better defined with alternative assay validation. For instance, sputum sample should also be analyzed using hemolysis assay. For the current data in Figure 5F, it is unclear if the GM \emptyset has a detection accuracy of 5 out of 7 (~70%) or if GM \emptyset is able to distinguish specific features associated with different clinical samples.

16. Are the GM \emptyset able to exclude PI indefinitely? Or is there a time window and preservation protocol for preserving GM \emptyset membrane integrity? Such information needs to be provided to convey clinical translationability of the system.

Minor points

17. For all data in this manuscript, authors need to describe their statistical method and whether all data are representative of at least three independent experiments in some figure legends (Fig. 3E, Fig. 4I). Before choosing method, they should test whether their data is normally distributed before deciding what statistical test to carry out.

18. IRB protocol number is missing for the use of human sputum.

19. Bacteria number in Figure 5F is strange. How are the authors able to detect low bacteria number in the single digit in clinical samples. It is likely a mislabel.

Reviewer #2 (Remarks to the Author):

This work describes a novel experimental approach to improve pathogen detection by exploiting the ability of macrophages to capture pathogens. Macrophages (cell line RAW264.7 cells) are made resistant to harsh handling conditions through intracellular hydrogelation, and are loaded with pathogen-specific functional nucleic acids. Further, internalisation of magnetic nanoparticles by macrophages facilitates enrichment using a magnet.

These cells are used to detect pathogens and toxins. Toxins are detected using propidium iodine (PI) that only penetrates damaged cells.

I won't comment on the efficiency of pathogen detection compared to other methodologies or the methodology used to incorporate hydrogels into the cells as they are out of my area of expertise.

General comments:

The authors focus on MR as an important pathogen-capture mechanism and exploit IL-4 to increase MR expression (Figure 1), which correlates with increased pathogen capture by IL-4-treated cells (Figure 2). Authors should consider that CD206 does not bind all pathogens and more extensive analysis of surface markers and how they are affected by IL-4 and the gelation process is required. Particularly because of the unusual pattern of bacteria binding shown in Figure 3 A (how representative are these images?). It is unclear if any bacteria has been internalised and there is no information on the mechanism of binding. Membrane (and MR) labelling should be used to delineate cellular membrane and endosomal compartment. Is bacteria binding inhibited by specific sugars? Does it require Calcium? Previous work with dendritic cells indicate that membrane features are maintained after the gelation process (references 24 and 25) and this was also confirmed by this work but as far as I know no endocytosis or phagocytosis after gelation has been demonstrated. Is it worth looking at this?

Technical/presentation queries:

What would happen if a non-macrophage cell line is used? Apart from lack of uptake of magnetic particles, which presumably requires phagocytosis, would bacteria attach to these cells after gelation?

There is no quantification of the efficiency of macrophage recovery after magnetic purification.

PI is routinely employed in flow cytometry. I would suggest testing toxicity using this technique, which will facilitate quantification.

Use of pathogen mutants lacking expression of toxins would strengthen the toxicity data.

Is candidalysin responsible of *C. albicans*-cytotoxicity? Could authors comment?

Figure 1F should include TEM of untreated macrophages.

The authors show the stability of the cells but, as far as I can tell, do not show if they maintain their pathogen binding activity. This would be essential if this technique is to be implemented in a clinical lab. Are expression of CD206 and binding activity of the cells reduced upon storage? Would they need to add additional IL-4?

IL-4 has been shown to regulate numerous markers in macrophages and act as cellular

survival factor. Are its effect just due to the fact that cells are just healthier?
It would be worth testing a series of cellular markers in the presence and absence of gelation in untreated and IL-4-treated cells.
Blocking Ab experiment in Figure S6 needs further explanation. There is no control antibody and, to my knowledge, neither E. coli or S. aureus bind MR. Further, S. aureus could bind antibodies through Fc portion due to protein A. Anti-MR antibodies used, do not seem validated for blocking its activity.
The experiments using antibody-coated particles is poorly described.
It is unclear how data in Figure 5F were generated. Is there any normalisation? Number of bacteria seem very low. According to supplementary information "After collecting GMØs using an external magnet, the adsorbed number and PI fluorescence of bacteria were measured with the help of a confocal microscope". Was this done by CFU or using S. aureus-specific DNAzyme?
Number of repeats in supplementary information and some figures in main document are missing.

Reviewer #3 (Remarks to the Author):

In this manuscript, the authors present a modified macrophage-based bacterial capture platform that incorporates DNAzyme decoration to allow for the detection of target bacterial strains. The basis of the proposed platform involves modifying macrophages via hydrogelation to create gelated macrophages (GMØ) with improved strength and stability. Magnetic nanoparticles were added to the GMØs to permit collection. The characterization and bacterial capture capabilities of the platform was evaluated via confocal microscopy with fluorescent staining, TEM, bacterial cultures, and flow cytometry. The developed GMØs exhibited no change in viability following the internalization of magnetic particles, and showed improved viability when exposed to a range of harsh environmental conditions. Bacterial assays indicated successful bacterial capture and specific detection following the incorporation of DNAzymes. Lastly, the authors also assessed pore-forming toxin detection from bacterial secretions through the dye-based monitoring of GMØ membrane destruction. I think the idea is very interesting and novel, however there major concerns to be addressed before the manuscript can be considered for publication in Nat. comm.:

- The concentration of internalized magnetic particles should be evaluated especially since the colour change is being used to confirm successful magnetization of the macrophages. Specifically, the optical properties of the solutions shown in Figure 1a should be quantified.
- Figure 2b should be reorganized to better convey trends between the three different macrophages. Statistical significance should also be included on the graph.
- Since photoinitiator is used to facilitate the crosslinking, addition of the fluorescent monomer without the photoinitiator should be assessed to confirm successful crosslinking.

- The intensities shown in Figure 2c should be quantified.
- What do the authors mean by differences in topography associated with increased bacterial aggregations around the cells? Is this indicative of increased MR presence or other changes in cell topography? If other changes are present, topographical characterization should be considered.
- Figure 3c and 3e should use log scale to better represent the differences between tested conditions.
- The data presented in Figures 3c show that the included control samples successfully capture a relatively high concentration of bacteria. How much of this bacterial capture is from insufficient washing, yielding transfer of uncaptured bacteria into the quantified test solution? If non-specific transfer is occurring, then these numbers are inaccurate. The authors should evaluate if the cell numbers presented in 3c decrease significantly with increased washing.
- Under a similar premise, the data presented in S6 should include unstimulated Mø as a control given that antibody-mediated blocking will be more effective against the innate level of MRs.
- The limit of detection should be experimentally derived.
- Moreover, the authors claim that their LOD is better than other previous works using the same DNAzyme. This claim should be better substantiated with mention of previously reported LODs.
- Given that the toxins discussed in Figure 5 are present within MRSA, why did the lysis of the GMø not influence the DNAzyme sensing results?
- Schematic Figure 5a should be more detailed to provide better context to the modifications performed.
- Confocal images obtained for 5f should be included.
- Tests with the patient samples should also evaluate DNAzyme signals and demonstrate a practical application.

Detailed Changes of the Manuscript and the Point-by-Point Response to the Reviewers' Comments

Reviewer #1

In this article “A Smart Pathogen Detector Engineered from Intracellular Hydrogelation of DNA-Decorated Macrophages”, the authors leverage a recently developed cell gelation technique to develop gelled macrophages (GMøs) for pathogen recognition. Three distinctive modalities are proposed: 1.) magnetic GMøs for bacteria enrichment. 2) DNzyme-conjugated GMøs for strain-specific bacteria detection. 3) A PI-based fluorescence approach for pore-forming toxin detection. Notably, the authors tested the PFT sensing modality of the GMø with sputum samples collected from *S. aureus*-caused pneumonia patients. The article is interesting in leveraging an emerging biotechnology for novel pathogen-sensing design. However, due to the three distinctive modalities proposed by the authors, the article is overextended and lacks sufficient depth and backgrounds for each of the proposed modality. In addition, characterizations of the GMøs, particularly elements related to bacterial engagement can be bolstered. Specific comments are as follows:

Major points

General info:

1. Due to the three distinctive modalities proposed by the authors, the background does not sufficiently describe the challenges each modality is aimed at tackling. To some extent the three distinctive modalities are antagonistic to each other. For instance, the latter two modalities have no need for magnetic extraction. How each modality would offer its diagnostic value and potentially complement each other should be described.

Response: Many thanks for the helpful suggestion. In our work, the first modality, i.e., magnetic separation and bacteria capture, play fundamental roles in our system. For example, a large number of impurities (e.g., somatic cells or fluorescent substances) exist in the biological samples, which severely disrupts the subsequent fluorescence analysis. After a magnetic separation step, the gelled cell particles can be easily collected and used for microscopic observation and flow cytometry analysis, which avoids the interference of other impurities and the loss of gelled cells during repeated washing/centrifuging steps. As a result, the operation difficulty is reduced, and detection performance is improved. Specific detection of bacterial cells or toxin analysis represent two main streamline directions in bacteria analysis, which can be used to evaluate the virulence of bacteria synergistically. In our revised manuscript, we add more description to claim the relationships between these modalities (Please see “*With this innovative tool in hand, we can achieve several goals: 1) GMøs have robust gelled cores and intact cell membranes, which allows them to resist different harsh conditions destructive to living cells. 2) GMøs pretreated with magnetic nanoparticles (MNPs) can be manipulated by an external magnet, so we can use an external magnet to conveniently separate bacteria-adsorbed cell particles from complex biological*

media with minimal loss. Since the existence of a large number of impurities in the biological samples, magnetic separation plays a fundamental role in the following analysis. 3) GMøs can efficiently recognize, capture, and enrich a broad spectrum of bacteria to their surface, and the close proximity distance between bacteria and the underlying cell membrane will easily activate sensing elements or facilitate the adsorption of bacteria-associated secretions on the cell membrane, which can dramatically improve the detection performance. 4) GMøs can be decorated with responsive DNA devices (e.g., DNzyme) for specific bacteria biosensing using a capture-and-detect strategy. Moreover, because of the unique structural characteristics of GMøs, they can be further applied to analyze bacteria-associated toxins, which not only provides more information about the virulence of the infected bacteria but also further extends the applicability of the GMø-based assay.”, **Page 3**).

2. The introduction should include descriptions regarding features and backgrounds of the gelated cell technology in recent references [Nat. Commun. 10, 1057, 2019][Adv. Mater. 33, 2101190, 2021]. In addition, as these priors work provide many of the evaluation techniques adopted in the present study (i.e., water test), proper citation should be added. Proper referencing of these prior work is critical to establish the foundation for the present study, which concerns primarily about new applications of the gelated cell system and has relatively shallow characterizations and discussion of the gelated cells' inherent properties.

Response: Many thanks for the helpful suggestions. In our revised manuscript, we add more description about these reports in the **Introduction** part (Please see “*Very recently, Hu and coworkers described a facile hydrogelation approach to assemble synthetic hydrogels inside the cells without disturbing the fluid and functional plasma membranes, making them suitable for subsequent biological application (e.g., ex vivo T-cell modulation)*^{24,25}. In this work, we apply this intracellular hydrogelation technique to transform Møs and the resulting gelated Møs (GMøs) can successfully overcome the abovementioned problems, making them suitable for in vitro use.” **Page 3**).

We also cited the references in the **Results** part (Please see “*The resultant GMøs exhibited discriminative membranous and hydrogel parts (Fig. 1c), suggesting the successful intracellular hydrogelation, which is in well agreement with the previous results*²⁵. Also, the water exposure assay could be used to rapidly evaluate the hydrogelation of Møs since living Møs rapidly ruptured under hypo-osmotic stress²⁵, whereas GMøs with robust crosslinked interiors remained intact without obvious morphology change (Fig. 1d and Supplementary Fig. 6).” , **Page 6**).

3. The specific choice of RAW264.7 should be described as opposed to alternative cellular options. Why choose a cell line of murine origin rather than that of human origin. One would expect that in clinical testing RAW264 cells are more susceptible to complement activation and membrane perforation in human serum.

Response: Many thanks for the helpful suggestions, so we have added more description about RAW264.7 in the revised manuscript (Please see “*Of note, we use RAW264.7 (a murine macrophage cell line) that is one of the most widely used immortalized macrophages for in vitro experiments as the basic cell material because of its convenient mass cultivation and easy maintenance without adding any inducers.*”, **Page 4**).

RAW 264.7 and human-derived THP-1 cells are two widely used model cell lines because of their immortalization, which can be easily produced in quantity. However, the use of THP-1 cells is not convenient because extra agents (e.g., phorbol myristate acetate, etc.) are needed to induce their differentiation to macrophages, which not only prolongs experiment time but also increases costs and operation complexity. In fact, the bacteria binding and capture capacity of THP-1 cell particles have also been tested in our preliminary experiments (data not shown), but their performance is slightly inferior to that of RAW264.7. So, in our proof-of-concept study, RAW264.7 cells are used as the basic material for constructing gelated cell particles.

For biosafety, most biological samples (e.g., serum) need extra pretreatments such as mild heat inactivation, centrifugation, or dilution before final analysis. After processing, the interference of complement proteins is negligible, and most of them lose activity or undergo degradation. In fact, we do not observe any performance changes of cell particles for bacteria capture in the complex media (Please see **Fig. 3e** and **Supplementary Figure 18**).

4. Mannose receptor is the sole membrane protein the authors highlight as a bacteria ligand. However, there are many other receptors and pattern recognition receptors (PRRs), such as Toll-like receptors, scavenger receptors, glycan receptors, on the surface of M ϕ that take part in bacterial recognition (Ref: Annu. Rev. Immunol. 23:901–44, 2005). In the context of the present work, these ligands relevant to bacterial binding should be characterized on the GM ϕ s. The present manuscript in fact has no membrane protein characterization of the GM ϕ s and only shows mannose receptors on M ϕ s before gelation in Figure 1B.

Response: Many thanks for the helpful suggestions. We chose MR as the main element for enhancing bacteria capture for the following reasons.

1) The main PRRs for recognizing bacteria components are TLRs, retinoic acid-inducible gene-I-like receptors (RLRs), nucleotide oligomerization domain-like receptors (NLRs), and C-type lectin receptors (CLRs). Among them, RLRs and NLRs exist in the cytoplasm, so they are precluded since the reactions in this study only occur on the cell membranes. Among various TLRs, three TLRs expressed on the membrane are well-known for recognizing different bacteria components. TLR2 is essential for the recognition of a variety of PAMPs from Gram-positive bacteria, including bacterial lipoproteins, lipomannans and lipoteichoic acids. TLR4 predominantly binds to lipopolysaccharides that are widely present on Gram-negative bacteria, and TLR5 mainly detects bacterial flagellin. Previous reports have used different cell extracts from Gram-positive & Gram-negative bacteria with an optimized ratio for macrophage

stimulation (Reference: Pretreated Macrophage-Membrane-Coated Gold Nanocages for Precise Drug Delivery for Treatment of Bacterial Infections, *Adv. Mater.*, 2018, 1804023), which is complex and cumbersome. Moreover, the enhancement of TLR2/4 expression on the macrophage membrane is limited (e.g., 1~2 folds). So, in this paper, we don't choose this method to activate macrophages.

2) In this work, we aim to use IL-4, a well-known cytokine that can transform M0-type macrophages into M2-type macrophages, which have the unique membrane markers such as CD115, CD163, CD206, and CD209. Among them, only CD163 (i.e., scavenger receptor) and CD206 (i.e., mannose receptor) participate in bacteria recognition and binding; meanwhile, their expression is significantly up-regulated compared with other receptors, further highlighting their potential roles. The MR is a C-type lectin, which is able to recognize a wide range of Gram-negative and Gram-positive bacteria, yeasts, parasites and mycobacteria (Reference: The mannose receptor is a pattern recognition receptor involved in host defense, *Curr. Opin. Immunol.*, 1998, 10, 50-55); CD163 also has a broad range of binding ligands including proteins, polyribonucleotides, polysaccharides and lipids for which the main common feature is that they are "polyanionic". Although the main function of CD163 is to the binding of Hemoglobin:Haptoglobin (Hb-Hp) complexes (Identification of the haemoglobin scavenger receptor, *Nature*, 2001, 409, 198-201), some reports reveal that CD163 also contributes to the bacteria binding.

Based on the above arguments, we investigate the expression of CD206, CD163, TLR2, and TLR4 on the Mø_s before and after IL-4 stimulation in our revised manuscript, and add more description about the reason to use mannose receptor and scavenger receptor as the main objects for bacteria capture (Please see "*After treatment with IL-4, M0-type Mø_s are transformed into M2-type Mø_s with upregulation of some unique membrane markers such as CD115, CD163, CD206, and CD209. Among them, only CD163 (i.e., scavenger receptor, SR) and CD206 (i.e., mannose receptor, MR) directly participate in bacteria recognition and binding^{17, 26}, so their expression on the membrane was evaluated using flow cytometry analysis. As a comparison, other potential markers such as TLR2 and TLR4 that can bind to bacterial components were also tested. The expression of CD163 and CD206 but not TLR2 and TLR4 was significantly promoted using IL-4 stimulation (Fig. 1b and Supplementary Fig. 5), consistent with previous reports^{22, 27}. These results suggested the potential role of CD206 and CD163 in bacteria capture (vide infra).*", **Page 6** and revised **Fig. 1b**).

5. The introduction of DNA duplex in Figure 2 and the adoption of DNA-GMø_s for bacteria capture in Figure 3 are very confusing. The way the DNA is introduced makes it appear as a key component for bacteria binding. Upon close scrutiny, however, the DNA only serves in the DNAzyme detection modality, which makes its appearance in Figure 3 regarding bacteria capture unnecessary and distracting. The data reorganized for better clarity.

Response: Many thanks for the helpful suggestions. The results in Fig. 2 mainly talk about the stability of GMø_s which not only involves cell particles but also the key

molecules on the membranes (e.g., DNA and protein markers). In Fig. 3, we have mainly tested the capture capacity of GMØs without DNA modification. We also demonstrate that the modification of DNA molecules on the cell particles does not affect the bacteria binding, which is important for subsequent bacteria analysis. To avoid this misunderstanding, we correct the title of Fig. 3 (Please see “*Recognition and capture capacity of GMØs for different microbes.*”).

Bacteria Capture modality:

6. The capture modality in Figure 3 only shows qualitatively the differences between GMØs with and without IL-4 stimulation. Yet there is no quantitative evaluation regarding the GMØs’ capture efficiency. For instance, a titration study with varying GMØs is important in assessing the affinity of the system and provide a picture on the number of GMØs one intends to use in a given volume of clinical sample (i.e. blood or urine). In addition, improvement of detection limit before and after designated GMØs-mediated enrichment should be demonstrated. The authors can refer to prior references for experimental designs aimed at demonstrating pathogen enrichment for enhanced detection (ACS Appl. Mater. Interfaces 2017, 9, 39953–39961).

Response: Many thanks for the helpful suggestions, so we have done a titration study in the revised manuscript. To do so, we use GFP-expressing bacteria and fix bacteria number (10^5 cells) in the solution and then catch them using different concentrations of GMØs. Thus, the capture efficiency can be facily evaluated by fluorescence change in the solution (Please see **Supplementary Figure 13**), which further demonstrates the advantage of the stimulated GMØs for bacteria capture.

Supplementary Figure 13. Titration assay showing the capture efficiency of unstimulated or IL-4-stimulated GMØs for (a) *E. coli* and (b) *S. aureus* capture. The number of GFP-expressing bacteria in solution is fixed to be about 10^5 cells. Fluorescence spectra showing the fluorescence change of bacteria solution before and after incubating with stimulated GMØs (10^4 GMØs). The error bars represent mean \pm SEM, n = 3.

We also use the recommended method to evaluate the advantage of magnetic enrichment. To do so, we have spiked *E. coli* (10^4 CFU mL⁻¹) into blood samples and performed the detection with or without magnetic enrichment. As shown in **Supplementary Figure 19**, bacteria-enriched samples showed an 8.6-fold enhancement in fluorescence. Without magnetic enrichment, a large number of impurities such as blood cells, or cell debris in the samples will dramatically influence the flow cytometry analysis and mask the positive signal, which causes a low signal-to-noise ratio. We also add corresponding description about this result (Please see “*The role of magnetic enrichment in bacteria detection was also verified by detecting blood samples spiked with a low-concentration E. coli (10⁴ CFU mL⁻¹). As shown in Supplementary Fig. 19, bacteria-enriched samples showed an 8.6-fold enhancement in fluorescence, thus verifying the advantageous role of the magnetic enrichment step in bacteria analysis. In addition, a large number of impurities that exist in the biological samples (e.g., exfoliated cells, blood cells or cell debris) dramatically disrupt the analysis, while a simple magnetic enrichment and separation step can address these problems³⁴.*”, **Page 13**).

Supplementary Figure 19. Positive role of magnetic enrichment in *E. coli* detection. A low-concentration *E. coli* (10^4 CFU mL⁻¹) was spiked in the 0.1% blood samples. After mixing the Dz^{EC}-GMØs (10^5 particles) with bacteria-containing blood samples, magnetic enrichment process was performed and the cell particles were transferred into a new tube and incubated for 30 min. Then, the cell particles were analyzed by flow cytometry analysis. As a comparison, no enrichment process was performed and the samples were directly measured by flow cytometry. The error bars represent mean \pm SEM, n = 3.

7. Comparison between GMØs and Ab-conjugated magnetic beads in figure 3E needs

proper standardization. Gelated cells and magnetic beads have significantly different surface area, density, and surface ligand number. Proper standardization with sound justification are needed for relevant comparison.

Response: Many thanks for the helpful suggestions. In our manuscript, we have chosen beads with a diameter of 10 μm , which is comparable to the size of GM ϕ s. As for the surface ligand density, it is difficult to strictly standardize the surface ligand density between GM ϕ s and Ab-conjugated beads since an array of ligands on the GM ϕ s contribute to the bacteria recognition and binding. We optimize the antibody density ($\sim 5.8 \times 10^5$ molecules/bead) through adjusting the modification concentration and conjugation time, which is comparable to the protein receptors on the cell membrane. In addition, we use the same experimental conditions to ensure comparability as possible.

DNAzyme detection modality

8. In Figure 4, the combination of GM ϕ inserted with specific-designed DNAzyme system for *E. coli* or *S. aureus* is promising with good sensitivity. However, as the design is aimed at improving a previously developed fluorogenic diagnosis technology, (Ref: *J Vis Exp.* (63): 3961., 2012), sensitivity improvement by the GM ϕ system over free DNAzyme should be demonstrated. It is presently unclear how much improvement is afforded by the GM ϕ system over the free DNAzyme approach.

Response: Many thanks again. In our system, we can either use flow cytometry to rapidly detect pathogens or observe the fluorescence response of a single GM ϕ after adding targets, making the detection method more flexible than traditional DNAzyme-based methods. Additionally, we have compared our method with previously reported methods for bacteria detection using DNAzymes (Please see **Supplementary Table 2**), which confirms the advantage of the proposed method.

9. The robustness of the DNAzyme detection modality should be demonstrated in physiologically relevant media in order to show the practical value of the device.

Response: Many thanks for the helpful suggestion, so we have used the Dz-GM ϕ s to detect *E. coli* spiked in different biological media such as serum, saliva, urine, and cell extracts (Please see “*Additionally, the performance of this method is not compromised by using complex biological samples such as serum, saliva, and urine (Supplementary Fig. 18).*” **Page 12**).

Supplementary Figure 18. Flow cytometry results of Dz^{EC}-GMØ-based assay for the detection of *E. coli* (10^6 CFU mL⁻¹) spiked in different biological media. The error bars represent mean \pm SEM, n = 3. MFI: mean fluorescence intensity.

PI-based PFT and pathogenic bacteria detection assay modality

10. The adoption of the GMØ for detecting membrane-damaging virulence factors is interesting, but the ambitious claims laid out by the authors require additional supporting data. Firstly, the proposed detection assays have existing alternatives using hemolytic assays for PFT and hemagglutination assays for membrane-binding bacteria. The sensitivity of the proposed system over these common approaches should be compared.

Response: Many thanks for the helpful suggestions. In our original manuscript, we have compared the GMØ-based assay with the commonly used hemolytic assay for toxin analysis (Please see the description: *Since the fluorescence signals at 10 fM were distinguishable from the background signals, the limit of detection (LOD) of the PI staining assay for Hla detection was estimated to be 10 fM (Fig. 5d, inset), which was significantly lower than the previously reported whole-cell-based hemolytic assay (nM range)³⁷, enzyme-linked immunosorbent assay (pM range)³⁸, fluorescence assay (pM range)³⁹ and field effect transistor sensors (nM)⁴⁰, further emphasizing the unique advantage of our approach for pathogen-associated toxin analysis. Page 15*). To further strengthen this, we have listed the performance to show the advantage of our method (Please see **Supplementary Table 3**).

11. A singular β -PFT is used in the present study, yet there are two major classes of PFTs, α -PFTs and β -PFTs, and different class families recognize their target host cells with high specificity (Ref: Nat Rev Microbiol. Feb;14(2):77-92, 2016). These PFTs recognize specific lipid constituents, sugar moieties, or protein receptors of the target membrane. Thus more types of PFTs should be included to define the applicability of the GMØ.

Response: Many thanks for the helpful suggestions. Since monocytes are the main targets for PFTs, we believe that the GMØ-based assay can detect a majority of PFTs.

Moreover, due to the existence of diverse cells, the applicability of our method can be guaranteed. In our revised manuscript, we also use flow cytometry to rapidly analyze another toxin, i.e., Cytolysin A (α -PFT), using the GM \emptyset -based assay (Please see **Supplementary Figure 24** and **corresponding description** “Finally, we tested the versatility of our GM \emptyset -based assay using another bacteria-associated α -PFT, i.e., cytolysin A (ClyA). To rapidly obtain fluorescence signal, we used flow cytometry analysis to measure the PI fluorescence of GM \emptyset s after incubating with different concentrations of ClyA, and results confirmed the successful quantitative analysis of ClyA (Supplementary Fig. 24).”, **Page 15**).

Supplementary Figure 24. GM \emptyset -based assay for analysis of cytolysin A (ClyA) using (a) Fluorescence microscopy images of the PI-stained GM \emptyset particles with or without ClyA treatment (1 μ M). Scale bar: 20 μ m. **b** Flow cytometry analysis of the PI-stained GM \emptyset particles incubated with different concentrations of ClyA. From bottom to up: 0, 10, 100, 10³, 10⁴, and 10⁵ nM. **c** Fluorescence intensities of the PI-stained GM \emptyset s incubated with different concentrations of ClyA. MFI: mean fluorescence intensity.

12. The depiction regarding Figure 5E is unclear. It is unclear whether the authors propose that the PI-based method can detect pathogenic bacteria because they can secrete PFTs or because they can damage cell membrane upon adherence. Such clarification is critical as it concerns whether the presence of bacteria is needed during sample analysis.

Response: Many thanks for the helpful suggestions, so we have added more description in the revised manuscript (Please see “Among these microorganisms, *E. coli* DH5a and *Saccharomyces cerevisiae* (*S. cerevisiae*) are mild strains that do not secrete hemolytic toxins, while *V. parahemolyticus*, *S. pyogenes*, *P. aeruginosa*, and *C. albicans* are known to secrete PFTs or directly damage cell membranes. For example, *V. parahemolyticus* secretes thermostable direct hemolysin (TDH) that can form pores in the membrane lipid bilayer⁴¹; *S. pyogenes* produces a membrane-damaging protein, i.e.,

streptolysin O and P. aeruginosa cooperatively use exolysin (ExlA) and Type IV Pili to exert its cytotoxic activity by promoting close contact between bacteria and the host cell^{42, 43}. As for C. albicans, it forms hyphae or secretes candidalysin to cause cell membrane damage⁴⁴.”, Page 16).

13. Quantitative analysis for pathogenic bacteria detection should be performed.

Response: Many thanks for the helpful suggestion. We have used flow cytometry to quantitatively detect *E. coli* and *S. aureus* (Please see **Fig. 4e** and **Supplementary Figure 21**).

14. If detection of pathogenic pathogen is based on PFT, specific PFTs for the pathogenic bacteria should be described and evaluated in parallel.

Response: Many thanks again. In our method, we have used DNAzymes to specifically detect pathogens (Please see **Fig. 4**), because DNAzymes can recognize certain pathogens and cleave their DNA substrates to produce fluorescent signals. In our PFT assay, we mainly show another application of the GMø for the analysis of bacteria secretions.

15. The context of the GMø-based diagnosis with patient sputum needs to be better defined with alternative assay validation. For instance, sputum sample should also be analyzed using hemolysis assay. For the current data in Figure 5F, it is unclear if the GMø has a detection accuracy of 5 out of 7 (~70%) or if GMø is able to distinguish specific features associated with different clinical samples.

Response: Many thanks for the helpful suggestions. All *S. aureus*-positive sputum samples have been confirmed and collected from a local hospital. In our revised manuscript, we further use a conventional hemolysis assay to evaluate the secretion of PFTs by bacteria in the sputum samples (Please see **Supplementary Figure 26**), which is in well agreement with our GMø-based assay.

16. Are the GMø able to exclude PI indefinitely? Or is there a time window and preservation protocol for preserving GMø membrane integrity? Such information needs to be provided to convey clinical translation ability of the system.

Response: Since the prepared GMø keep intact membrane structures, no PI molecules can permeate into the cells during our experiment (Please see **Fig. 5b**). In our revised manuscript, we stored the GMø in the serum-free cell cryopreservation solution at -20 °C for 2 weeks and no obvious PI staining of the GMø was observed (Please see **Supplementary Figure 23**), thus demonstrating the robustness of the cell particles.

Minor points

17. For all data in this manuscript, authors need to describe their statistical method and whether all data are representative of at least three independent experiments in some figure legends (Fig. 3E, Fig. 4I). Before choosing method, they should test whether their data is normally distributed before deciding what statistical test to carry out.

Response: Many thanks for the helpful suggestion, so we have added statistical annotations in the revised manuscript. In all figures, the error bars represent mean \pm SEM, n = 3.

18. IRB protocol number is missing for the use of human sputum.

Response: Many thanks for the helpful suggestion, so we have provided the IRB protocol number in the supporting information (Please See “*All sputum samples were acquired and handled according to the protocols approved by the Scientific Ethical Committee of the First Affiliated Hospital of Nanjing Medical University (No. 2021-SPFA-360)*”, **Page 21**).

19. Bacteria number in Figure 5F is strange. How are the authors able to detect low bacteria number in the single digit in clinical samples? It is likely a mislabel.

Response: Many thanks for the helpful suggestions. To eliminate the misunderstanding of Figure 5F, we replaced them with the unbiased flow cytometry analysis. In our revised manuscript, we used Dz^{SA}-GM ϕ -based assay to simultaneously detect both *S. aureus* and *S. aureus*-associated Hla (Please see **Fig. 5f** and corresponding description “*After a simple capture-and-detect step, the resulting Dz^{SA}-GM ϕ s were collected by an external magnetic field and subjected to flow cytometry analysis. In this assay, the presence of S. aureus in the samples would activate the DNAszymes, generating green fluorescence on the cell membranes, and Hla could punch GM ϕ s and cause an increase in PI fluorescence. As shown in Fig. 5f, all patient samples generated a positive signal for S. aureus (left panel), which was ascribed to the activation of DNAszymes on the gelated cell particles. Meanwhile, PI staining experiment (right panel) showed that not all S. aureus secreted Hla (i.e., P2 and P5), which was in agreement with the results of the conventional hemolytic assay (Supplementary Fig. 26). More details are shown in Supplementary Fig. 27. These results revealed that our method had the ability to rapidly analyze bacteria and toxin in clinical samples.*”, **Page 16**). The presence of *S. aureus* in the patient samples will activate the DNAszymes, generating green fluorescence on the cell membrane, and Hla will punch GM ϕ s and cause an increase in PI fluorescence. Detailed results are also presented in **Supplementary Figure 26 and Supplementary Figure 27** in the supporting information.

Reviewer #2

This work describes a novel experimental approach to improve pathogen detection by

exploiting the ability of macrophages to capture pathogens. Macrophages (cell line RAW264.7 cells) are made resistant to harsh handling conditions through intracellular hydrogelation, and are loaded with pathogen-specific functional nucleic acids. Further, internalisation of magnetic nanoparticles by macrophages facilitates enrichment using a magnet. These cells are used to detect pathogens and toxins. Toxins are detected using propidium iodine (PI) that only penetrates damaged cells. I won't comment of the efficiency of pathogen detection compared to other methodologies or the methodology used to incorporate hydrogels into the cells as they are out of my area of expertise.

General comments:

The authors focus on MR as an important pathogen-capture mechanism and exploit IL-4 to increase MR expression (Figure 1), which correlates with increased pathogen capture by IL-4-treated cells (Figure 2). Authors should consider that CD206 does not bind all pathogens and more extensive analysis of surface markers and how they are affected by IL-4 and the gelation process is required. Particularly because of the unusual pattern of bacteria binding shown in Figure 3 A (how representative are these images?).

Response: Many thanks for the helpful suggestions. We chose MR as the main marker for enhancing bacteria capture for the following reasons.

1) The main PRRs for recognizing bacteria components are TLRs, retinoic acid-inducible gene-I-like receptors (RLRs), nucleotide oligomerization domain-like receptors (NLRs), and C-type lectin receptors (CLRs). Among them, RLRs and NLRs exist in the cytoplasm, so they are precluded **since the reactions in this study only occur on the surface of cell membranes**. Among various TLRs, three TLRs expressed on the membrane are well-known for recognizing different bacteria components. TLR2 is essential for the recognition of a variety of PAMPs from Gram-positive bacteria, including bacterial lipoproteins, lipomannans and lipoteichoic acids. TLR4 predominantly binds to lipopolysaccharide that are widely present on Gram-negative bacteria, and TLR5 mainly detects bacterial flagellin. Previous reports have used cell extracts from Gram-positive & Gram-negative bacteria in an appropriate ratio for macrophage stimulation (Reference: Pretreated Macrophage-Membrane-Coated Gold Nanocages for Precise Drug Delivery for Treatment of Bacterial Infections, *Adv. Mater.*, 2018, 1804023), whose process is complex and cumbersome. More importantly, the upregulation of TLRs on the macrophage membrane is limited (e.g., 1~2 folds). So, we don't choose this method to activate macrophages.

2) In this work, we use IL-4, a well-known cytokine that can transform M0-type macrophages into M2-type macrophages that have the unique membrane markers such as CD115, CD163, CD206, and CD209 and so on. Among them, CD163 (i.e., scavenger receptor) and CD206 (i.e., mannose receptor) mainly participate in bacteria recognition and binding; meanwhile, their expression is significantly up-regulated compared with other receptors, further highlighting their potential functionality. The MR is a C-type lectin, which is able to recognize a wide range of Gram-negative and Gram-positive bacteria, yeasts, parasites and mycobacteria (Reference: The mannose receptor is a pattern recognition receptor involved in host defense, *Curr. Opin. Immunol.*, 1998, 10, 50-55); CD163 prefers to bind to polyanionic ligands including

proteins, polyribonucleotides, polysaccharides and lipids, and some reports also confirm that CD163 contributes to the bacteria binding.

Based on the above arguments, we have focused on MR and found its positive role in bacteria capture and binding. In our revised manuscript, we investigate the expression of CD206, CD163, TLR2, and TLR4 on the Mø before and after IL-4 stimulation and further discover the positive role of CD163 in bacteria capture. More description about the reasons to use mannose receptor (CD206) and scavenger receptor (CD163) for bacteria capture has been added (Please see “*After treatment with IL-4, M0-type Mø are transformed into M2-type Mø with upregulation of some unique membrane markers such as CD115, CD163, CD206, and CD209. Among them, only CD163 (i.e., scavenger receptor, SR) and CD206 (i.e., mannose receptor, MR) directly participate in bacteria recognition and binding*^{17, 26}, so their expression on the membrane was evaluated using flow cytometry analysis. As a comparison, other potential markers such as TLR2 and TLR4 that can bind to bacterial components were also tested. The expression of CD163 and CD206 but not TLR2 and TLR4 was significantly promoted using IL-4 stimulation (Fig. 1b and Supplementary Fig. 5), consistent with previous reports^{22, 27}. These results suggested the potential role of CD206 and CD163 in bacteria capture (vide infra).”, **Page 6** and revised **Fig. 1b**).

It is unclear if any bacteria have been internalised and there is no information on the mechanism of binding. Membrane (and MR) labelling should be used to delineate cellular membrane and endosomal compartment.

Response: Many thanks for the helpful suggestion. After an intracellular gelation process, the cytoplasm is solidified by the network of hydrogel and cells rapidly die because of the toxicity of UV light-induced radical reactions, which has been demonstrated by CCK-8 assay (Please see **Supplementary Figure 7**). Thus, bacterial cells can't be internalized by dead cell particles.

Is bacteria binding inhibited by specific sugars? Does it require Calcium? Previous work with dendritic cells indicates that membrane features are maintained after the gelation process (references 24 and 25) and this was also confirmed by this work but as far as I know no endocytosis or phagocytosis after gelation has been demonstrated. Is it worth looking at this?

Response: Only high concentration of monosaccharides, e.g., mannose or fructose (sub-mM~mM range), has shown certain inhibitory effect for bacteria capture, which was probably due to the weak affinity of receptors toward sugars. Calcium is an optional component in the solution, which can indeed enhance the interaction between GMø and bacteria.

Again, after gelation process, cells rapidly die due to the toxicity effect of UV light-induced radical reactions. Although the cell particles retain their intact structures, they lose metabolic activity and no endocytosis or phagocytosis behaviors will occur.

Technical/presentation queries:

What would happen if a non-macrophage cell line is used? Apart from lack of uptake of magnetic particles, which presumably requires phagocytosis, would bacteria attach to these cells after gelation?

Response: In our revised manuscript, we have prepared another gelated cell, e.g., MCF-7 (human breast cancer cell line), which is not an immune cell and does not efficiently recognize bacteria. After capturing and washing, few bacterial cells can be observed around the MCF-7-based gelated particles (Please see **Supplementary Figure 12**). Thus, we can infer the indispensable role of macrophages as the basic scaffold material.

Supplementary Figure 12. Confocal images of gelated MCF-7 cells after incubation with *E. coli* and *S. aureus*. Scale bars: 5 μ m. Nuclei are shown in blue. The arrows show the location of the bacteria.

There is no quantification of the efficiency of macrophage recovery after magnetic purification.

Response: Many thanks for the helpful suggestion, so we have counted the number of cell particles before and after magnetic separation. Negligible cell loss (cell recovery \approx 98%) is seen after magnetic purification even after repeated operations, indicating the high recovery of GM ϕ s. We have described this result in the revised main text (Please see “As shown in Fig. 1a, the resulting cells (MNPs/M ϕ s) became dark compared to natural M ϕ s and could rapidly respond to an external magnet with a high cell recovery (\approx 98%).”, **Page 5**).

PI is routinely employed in flow cytometry. I would suggest testing toxicity using this technique, which will facilitate quantification.

Response: Many thanks for the helpful suggestions. We acknowledge that the PI signal can be facilely measured by flow cytometry analysis. In our original manuscript, we have used flow cytometry to quantitatively detect *E. coli* and *S. aureus*. Additionally, we further used flow cytometry analysis to measure another toxin, i.e., Cytolysin A (Please see **Supplementary Figure 24**).

In our work, although the single-cell-based fluorescence microscopy analysis is relatively cumbersome, it can provide better sensitivity since only one cell particle is needed to be punched by toxins and detected by fluorescent analysis. Alternatively, if one wants to rapidly screen the toxin in samples, they can use flow cytometry analysis with a moderate detection limit.

Use of pathogen mutants lacking expression of toxins would strengthen the toxicity data.

Response: Many thanks for the helpful suggestion. In our original paper, we have already tested two nonpathogen strains that do not express toxins (e.g., yeast and *E. coli* DH5 α). As anticipated, no PI staining of the GM ϕ s was found (Fig. 5e), confirming that the GM ϕ s can be used to evaluate the toxicity of bacteria. In our revised manuscript, since the *hla* mutant strain is not easily obtained, we use neutralizing antibody to verify Hl α -induced PI influx. As shown in **Supplementary Figure 22**, the PI fluorescence significantly reduced after introducing anti-Hl α antibody but not anti-CD115 antibody, which confirms that confirm the PI signal was due to the Hl α attack.

Supplementary Figure 22. **a** Flow cytometry analysis and **(b)** mean fluorescence intensity of the GM ϕ s with different treatments and subsequently stained by PI molecules. The error bars represent mean \pm SEM, n = 3. Statistical analysis was performed using one-way analysis of variance (ANOVA).

Is candidalysin responsible of *C. albicans*-cytotoxicity? Could authors comment?

Response: We have obtained pure candidalysin and used GM ϕ s to examine its pore-forming capacity. As shown below, fluorescence microscopy results show that the PI fluorescence increases with the increase of candidalysin concentration (Fig a). Notably, we think its pore-forming capacity is low compared with other common PFTs (e.g., Hl α and ClyA) since even a high concentration of candidalysin (e.g., 20 μ M) only results in moderate PI fluorescence enhancement (Fig a, b). We think that it is difficult to reach this concentration in the solution since our reaction only takes less than 2 h, so other factors (e.g., hypha puncture) may play a leading role in *C. albicans*-induced PI influx and candidalysin is a synergistic factor during this process.

(a) PI fluorescence images and (b) flow cytometry analysis of GMØs after incubating different concentrations of candidalysin (from left to right: 0, 1, 2, 5, 10, and 20 µM).

Figure 1F should include TEM of untreated macrophages.

Response: Many thanks for the helpful suggestion, so we have added a TEM image of untreated macrophages in the revised Fig. 1e.

The authors show the stability of the cells but, as far as I can tell, do not show if they maintain their pathogen binding activity. This would be essential if this technique is to be implemented in a clinical lab. Are expression of CD206 and binding activity of the cells reduced upon storage? Would they need to add additional IL-4?

Response: Many thanks for the helpful suggestion. In our revised manuscript, we also tested the pathogen binding activity of the GMØs after long-term storage (1 month). The plate culture assay demonstrates that the bacteria capture capacity of GMØs was hardly influenced (Please see **Supplementary Figure 15**), thereby verifying the robustness of the fabricated particles. In parallel, we also use flow cytometry analysis to evaluate the amount of CD206 and CD163 on the cell membrane. As shown in **Supplementary Figure 15**, no obvious reduction in fluorescence was observed, indicating that both markers remain stable on the membrane surface.

After gelation, the cells rapidly die (Please see CCK-8 assay, **Supplementary Figure 7**), so the markers on the cell membrane will not change after adding IL-4.

Supplementary Figure 15. Comparison of GMØs for (a) *E. coli* and (b) *S. aureus* capture before and after being stored for 1 month. c No obvious change was seen for CD206 (left) and CD163 (right) markers determined by flow cytometry analysis after storing the GMØs for 1 month. The error bars represent mean \pm SEM, n = 3. Statistical analysis was performed using one-way analysis of variance (ANOVA).

IL-4 has been shown to regulate numerous markers in macrophages and act as cellular survival factor. Are its effect just due to the fact that cells are just healthier?

Response: Many thanks for the helpful suggestion. After intracellular gelation, cells lose activity and finally die. However, because of the intact of cell membrane structures, the surface PRRs can still maintain their function for a long period. As a result, the recognition and binding capacity of GMØs for bacteria are not disrupted. Overall, IL-4 is used to stimulate the upregulation of certain PRRs on the cell membrane, and the enhanced capture effect is not due to the healthier cells.

It would be worth testing a series of cellular markers in the presence and absence of gelation in untreated and IL-4-treated cells.

Response: Many thanks for the helpful suggestion. In this paper, we mainly focus on the reaction on the cell membrane, which is directly correlated with bacteria recognition and binding in vitro. So, the change of cytoplasm markers is beyond our research scope. In our revised manuscript, we investigate the expression of CD206, CD163, TLR2, and TLR4 on the MØs before and after IL-4 stimulation since these receptors are directly correlated with bacteria recognition. More description about the reasons to use mannose receptor and scavenger receptor as the main markers for bacteria capture is added (Please see “*After treatment with IL-4, M0-type MØs are transformed into M2-type MØs with upregulation of some unique membrane markers such as CD115, CD163, CD206, and CD209. Among them, only CD163 (i.e., scavenger receptor, SR) and CD206 (i.e.,*

mannose receptor, MR) directly participate in bacteria recognition and binding^{17, 26}, so their expression on the membrane was evaluated using flow cytometry analysis. As a comparison, other potential markers such as TLR2 and TLR4 that can bind to bacterial components were also tested. The expression of CD163 and CD206 but not TLR2 and TLR4 was significantly promoted using IL-4 stimulation (Fig. 1b and Supplementary Fig. 5), consistent with previous reports^{22, 27}. These results suggested the potential role of CD206 and CD163 in bacteria capture (vide infra).”, Page 6).

Blocking Ab experiment in Figure S6 needs further explanation. There is no control antibody and, to my knowledge, neither *E. coli* or *S. aureus* bind MR. Further, *S. aureus* could bring antibodies through Fc portion due to protein A. Anti-MR antibodies used, do not seem validated for blocking its activity.

Response: Many thanks again. In our revised manuscript, we have added a control antibody targeting CD115 in the blocking experiment. CD115, also known as colony-stimulating factor 1 receptor, is another surface marker of M2 macrophages and is not directly responsible for bacteria binding. As shown in the revised **Supplementary Figure 14**, the capture efficiency of both unstimulated and stimulated cells remains unchanged when the control antibody is used.

The MR is a 180 kDa transmembrane protein that has five domains: the amino-terminal cysteine-rich region, which shares homology with ricin B chain; a domain containing a fibronectin type II repeat; a series of eight tandem lectin-like carbohydrate recognition domains (CRDs); a transmembrane domain; and a cytoplasmic carboxy-terminal domain (Reference: The mannose receptor is a pattern recognition receptor involved in host defense, *Curr. Opin. Immunol.*, 1998, 10, 50-55). Among them, CRDs recognize mannose, fucose, or N-acetylglucosamine residues and glycoproteins bearing sulfated sugars are recognized by the cysteine-rich domain. The presence of a large number of sugar ligands on the bacteria surface (e.g., *E. coli* and *S. aureus*) suggests the strong correlation between bacteria binding and MRs; moreover, some reports have also described the interaction between bacteria and MRs (Adherence of *Escherichia coli* to human mucosal cells mediated by mannose receptors, *Nature*, 1977, 265, 623-625; Stable Expression and Characterization of an Optimized Mannose Receptor, *J. Clin. Cell Immunol.* 2015, 6, 330; Identification and functional analysis of Mannose receptor in Asian swamp eel (*Monopterus albus*) in response to bacterial infection, *Fish Shellfish Immun.*, 2022, 127, 463-473). Moreover, in our revised manuscript, we also reveal the positive role of CD163 (i.e., scavenger receptor) in bacteria capture due to its strong binding to polyanionic ligands, and most bacteria are known to be negatively charged.

Supplementary Figure 14. Capture of (a) *E. coli* and (b) *S. aureus* using IL-4-unstimulated/stimulated GMØs with or without pre-blockage of different antibodies. Among them, anti-CD115 antibody is used as a negative control since CD115 is not involved in bacterial recognition. *S. aureus* was pre-treated with Fc fragments to avoid the interference of protein A on the bacterial surface. The error bars represent mean \pm SEM, $n = 3$. Statistical analysis was performed using one-way analysis of variance (ANOVA).

We acknowledge that the expression of Protein A by *S. aureus* is beyond our consideration; however, it just matches well with our previous results since we were surprised that the capture of *S. aureus* is less affected by the antibody blocking process compared to *E. coli*. In our revised manuscript, *S. aureus* pretreated with Fc fragments is used and an obvious inhibitory effect is observed.

The experiments using antibody-coated particles is poorly described.

Response: Many thanks for the helpful suggestions, so we have provided more details about the bacteria capture experiment using antibody-coated particles. (Please see “*To capture bacteria, 50 μ L of Ab-SiMBs (1×10^5 particles) solution was incubated with 50 μ L of bacteria suspension (final OD_{600 nm} = 0.2-0.3) for 15 min at room temperature with continuous shaking (350 rpm). Then, the Ab-SiMBs were separated by an external magnet and washed with PBS twice. After that, 5 μ L of Ab-SiMB solution was subjected to confocal microscopy analysis (Carl Zeiss LSM880, Germany).*”, **Page 20**).

It is unclear how data in Figure 5F were generated. Is there any normalisation? Number of bacteria seem very low. According to supplementary information “After collecting GMØs using an external magnet, the adsorbed number and PI fluorescence of bacteria were measured with the help of a confocal microscope”. Was this done by CFU or using *S. aureus*-specific DNAzyme?

Response: Many thanks for the helpful suggestions. To eliminate the misunderstanding of Fig. 5f, we replaced them with the unbiased flow cytometry analysis. In our revised manuscript, we used Dz^{SA}-GMØ-based assay to simultaneously detect both *S. aureus* and Hla (Please see **revised Fig. 5f** and **corresponding description** “After a simple capture-and-detect step, the resulting Dz^{SA}-GMØs were collected by an external magnetic field and subjected to flow cytometry analysis. In this assay, the presence of

S. aureus in the samples would activate the DNazymes, generating green fluorescence on the cell membranes, and Hla could punch GMø and cause an increase in PI fluorescence. As shown in Fig. 5f, all patient samples generated a positive signal for *S. aureus* (left panel), which was ascribed to the activation of DNazymes on the gelated cell particles. Meanwhile, PI staining experiment (right panel) showed that not all *S. aureus* secreted Hla (i.e., P2 and P5), which was in agreement with the results of the conventional hemolytic assay (Supplementary Fig. 26). More details are shown in Supplementary Fig. 27. These results revealed that our method had the ability to rapidly analyze bacteria and toxin in clinical samples.”, **Page 16**). The presence of *S. aureus* in the patient samples will activate the DNazymes, generating green fluorescence on the cell membrane, and Hla will punch GMø and cause an increase in PI fluorescence. Detailed results are also presented in **Supplementary Figure 26** and **27** in the supporting information.

Number of repeats in supplementary information and some figures in main document are missing.

Response: Many thanks for the helpful suggestions, so we have added this information in the revised figures (Please see Fig. 5, Supplementary Figure 3, 9, 14, 17, and 21). Error bars in all figures represented the standard error derived from three independent experiments.

Reviewer #3

In this manuscript, the authors present a modified macrophage-based bacterial capture platform that incorporates DNzyme decoration to allow for the detection of target bacterial strains. The basis of the proposed platform involves modifying macrophages via hydrogelation to create gelated macrophages (GMø) with improved strength and stability. Magnetic nanoparticles were added to the GMø to permit collection. The characterization and bacterial capture capabilities of the platform was evaluated via confocal microscopy with fluorescent staining, TEM, bacterial cultures, and flow cytometry. The developed GMø exhibited no change in viability following the internalization of magnetic particles, and showed improved viability when exposed to a range of harsh environmental conditions. Bacterial assays indicated successful bacterial capture and specific detection following the incorporation of DNazymes. Lastly, the authors also assessed pore-forming toxin detection from bacterial secretions through the dye-based monitoring of GMø membrane destruction. I think the idea is very interesting and novel, however there major concerns to be addressed before the manuscript can be considered for publication in Nat. comm.:

- The concentration of internalized magnetic particles should be evaluated especially since the color change is being used to confirm successful magnetization of the

macrophages. Specifically, the optical properties of the solutions shown in Figure 1a should be quantified.

Response: Many thanks for the helpful suggestions. In our original work, transmission electron microscopy images also confirm the internalization of magnetic nanoparticles in the cells (Fig. 1e). In our revised manuscript, inductively coupled plasma mass spectrometry (ICP-MS) is used to determine the internalization of magnetic particles in cells at different times (**Supplementary Figure 4**). Moreover, we use UV/vis absorption spectra to characterize GMø_s, MNPs, and MNPs/GMø_s samples. (Please see **Supplementary Figure 2**)

Supplementary Figure 2. Uv/vis spectra of GMø_s, MNPs, and MNPs/GMø_s. The absorption characteristic peak of MNPs is located around 310-400 nm, and large cell particles don't show any characteristic peaks. The presence of MNPs peaks suggests the successful internalization of MNPs by Mø_s.

Supplementary Figure 4. ICP-MS analysis of MNPs internalization at various times. The error bars represent mean ± SEM, n = 3.

• Figure 2b should be reorganized to better convey trends between the three different macrophages. Statistical significance should also be included on the graph.

Response: Many thanks for the helpful suggestions, so we have reorganized Fig. 2b and added statistical significance in the revised manuscript.

- Since photoinitiator is used to facilitate the crosslinking, addition of the fluorescent monomer without the photoinitiator should be assessed to confirm successful crosslinking.

Response: Many thanks again. Because of the direct permeation of the monomers across cell membrane, it is not easy to distinguish the successful gelation of cells by only adding the fluorescent monomers. According to previous reports (Nat. Commun. 10, 1057, 2019; Adv. Mater. 33, 2101190, 2021), a simple water test, which exposes cells to hypo-osmotic stress, has been used to evaluate the successful intracellular hydrogel crosslinking since non-crosslinked cytosols undergo rapid disintegration and fragmentation whereas gelled cells with crosslinked cores retain their overall structure in the presence of osmotic stress.

To do so, we performed a comparative study by incubating cells treated with (i) photoinitiator, (ii) monomer, (iii) photoinitiator + monomer, and (iv) photoinitiator + monomer + UV light. As anticipated, only cells in group iv survived after challenging with pure water (Please see **Supplementary Figure 6**), further confirming the successful intracellular gelation.

Supplementary Figure 6. **a** Bright-field microscopy observations and **(b)** statistical analysis of the Mø cells after different treatments upon suspension in pure water. i) photoinitiator, (ii) monomer, (iii) photoinitiator + monomer, and (iv) photoinitiator + monomer + UV light. The error bars represent mean \pm SEM, n = 3.

- The intensities shown in Figure 2c should be quantified.

Response: Many thanks for the helpful suggestion, so we have used flow cytometry to quantify the DNA change on the GMø cells after long-term storage (Please see revised Fig. 2c).

- What do the authors mean by differences in topography associated with increased bacterial aggregations around the cells? Is this indicative of increased MR presence or other changes in cell topography? If other changes are present, topographical characterization should be considered.

Response: Compared with synthetic beads with smooth surface (Please see below), the prepared GMø cells remain the cell-specific microstructures such as microvilli, wrinkles, and filopodias and so on, which have an improved surface area. As a result, the contact

chance between the cell particles and bacteria was enhanced, resulting in better recognition and capture efficiency (Advanced Materials, 2015, 27, 310-313) compared to conventional magnetic beads with smooth surfaces. In our revised manuscript, we used scanning electron microscopy (SEM) to confirm the topological structure of GMø (Please see **Supplementary Figure 10**).

SEM images showing the different surface morphologies of silica bead and GMø. Scale bar: 2 μm .

- Figure 3c and 3e should use log scale to better represent the differences between tested conditions.

Response: Many thanks for the helpful suggestion, so we have used log scale in the revised Fig. 3c and 3e.

- The data presented in Figures 3c show that the included control samples successfully capture a relatively high concentration of bacteria. How much of this bacterial capture is from insufficient washing, yielding transfer of uncaptured bacteria into the quantified test solution? If non-specific transfer is occurring, then these numbers are inaccurate. The authors should evaluate if the cell numbers presented in 3c decrease significantly with increased washing.

Response: Many thanks again. After bacteria capture, the cell particles are thoroughly washed for three times. Of note, the relatively high concentration of bacteria in the control groups (Fig. 3c) was not due to the unspecific adsorption between bacteria and cell particles. Native Mø also have the binding capacity for bacteria since they are immune cells that already have a certain amount of pattern recognition receptors even without any stimulation. As a result, Mø without IL-4 stimulation can still recognize various bacteria, causing the relatively high background counts in Fig. 3c. In this part, we mainly reveal that the stimulation of Mø by IL-4 can further improve their ability for bacteria recognition and capture.

In our revised manuscript, we have also prepared another gelated cell, e.g., MCF-7 (human breast cancer cell line), which is not an immune cell and does not efficiently bind and capture bacteria. After capturing and washing, few bacterial cells can be observed around the MCF-7-based gelated particles (Please see **Supplementary Figure 12**). Thus, we can infer that most bacteria can be removed from the cell particles after thoroughly washing and the nonspecific adsorption of bacteria on the cell surface

is negligible.

Supplementary Figure S12. Confocal images of gelated MCF-7 cells after incubation with *E. coli* and *S. aureus*. Scale bars: 5 μm . Nuclei are shown in blue. The arrows show the location of the bacteria. Of note, most bacteria did not adsorb on the surface MCF-7-based cell particles.

- Under a similar premise, the data presented in S6 should include unstimulated M ϕ s as a control given that antibody-mediated blocking will be more effective against the innate level of MRs.

Response: Many thanks again, so we have also tested the capture efficiency of the unstimulated M ϕ s before and after MR antibody blocking. From revised Supplementary Figure 14, we found that the capture efficiency of the unstimulated M ϕ s is less affected by antibody blocking. Quiescent macrophages have a relatively low-level PRRs on cell membranes, so the blocking of a limited number of receptors has less influence on bacteria capture. After proper activation, the levels of some PRRs (e.g., mannose receptor and scavenger receptor) are dramatically up-regulated, so the capture efficiency of the cell particles is significantly disturbed by these antibodies.

Supplementary Figure 14. Capture of (a) *E. coli* and (b) *S. aureus* using IL-4-unstimulated/stimulated GM ϕ s with or without pre-blockage of different antibodies. Among them, anti-CD115 antibody is used as a negative control since CD115 is not involved in bacterial recognition. *S. aureus* was pre-treated with Fc fragments to avoid the interference of protein A on the bacterial surface. The error bars represent mean \pm SEM, $n = 3$. Statistical

analysis was performed using one-way analysis of variance (ANOVA).

- The limit of detection should be experimentally derived.

Response: Many thanks for the helpful suggestion, so we have changed the method to determine the limit of detection through experimental data (Please see “*Since the fluorescence signals at 500 CFU mL⁻¹ were distinguishable from those at 0 CFU mL⁻¹, the limit of detection (LOD) of the Dz^{EC}-GMØ-based assay for E. coli detection was estimated to be 500 CFU mL⁻¹ (Fig. 4e, inset).*”, **Page 12**; “*Since the fluorescence signals at 10 fM were distinguishable from the background signals, the limit of detection (LOD) of the PI staining assay for Hla detection was estimated to be 10 fM (Fig. 5d, inset).*”, **Page 15**).

- Moreover, the authors claim that their LOD is better than other previous works using the same DNAzyme. This claim should be better substantiated with mention of previously reported LODs.

Response: Many thanks for the helpful suggestion, so we have compared our method with previously reported works (Please see Supplementary Table 2), which verifies the advantages of our method.

- Given that the toxins discussed in Figure 5 are present within MRSA, why did the lysis of the GMØs not influence the DNAzyme sensing results?

Response: For normal cells, toxin-induced pore formation disrupts the permeability barrier of the plasma membrane, eventually resulting in the lysis of cells. However, for gelled cells, the cytoplasm has been solidified by the network of hydrogel, and the change of osmotic pressure can't affect them (Please see **Fig. 1d** and **Fig. 2**). For example, the morphology and number of GMØs exposed in the pure water remain unchanged (**Fig. 1d**). Thus, the toxins adsorbed on the membrane hardly affect the integrity of GMØs and DNAzymes still exist on the cell membrane.

- Schematic Figure 5a should be more detailed to provide better context to the modifications performed.

Response: Many thanks for the helpful suggestion, so we have revised Fig. 5a and provided more information on this scheme.

Fig. 5a. Schematic illustration of GMØ-based method for PFT analysis.

- Confocal images obtained for 5f should be included.

Response: Many thanks for the helpful suggestion. Considering the confusing data presented in Fig. 5f, we replaced them with the unbiased flow cytometry analysis. Flow cytometry result of each sample has been presented in **Supplementary Figure 27** in the supporting information.

- Tests with the patient samples should also evaluate DNAzyme signals and demonstrate a practical application.

Response: Many thanks for the helpful suggestion. In our revised manuscript, we have used Dz^{SA}-GMØ-based assay to simultaneously detect both *S. aureus* and H1α. The presence of *S. aureus* in the patient samples will activate DNAzymes on the GMØs, resulting in an increased fluorescein signal (Please see **Fig. 5f**, left panel).

REVIEWER COMMENTS

Reviewer #1 (Remarks to the Author):

The authors have positively addressed all my prior comments. The revised manuscript has substantial data to support the novel cell-based construct for bacterial detection. The clinical study further supports the translational viability of the system. The authors are congratulated for a study well done. Some proreading should be performed prior to publication. For instance line 271 on page 10 "which was significantly than that" should be "which was significantly lower than that". Also Line 289 on page 10 "proofing" should be "proving".

Reviewer #2 (Remarks to the Author):

The authors have addressed multiple of my queries and the manuscript has been improved. Nevertheless there are some aspects that require clarification and revision.

Main manuscript

Introduction:

"After treatment with IL-4, M0-type M ϕ s are transformed into M2-type M ϕ s with upregulation of some unique membrane markers such as CD115, CD163, CD206, and CD209". This section needs a reference. CD209 is also able to recognize bacteria but CD209 is not expressed in mouse cells. As far as I know CD115 is not a classic marker of M2 macrophages.

Results section:

Data in Figure 1b requires isotype controls and the differences between "with and without IL-4" would be clearer if histograms were shown as overlays.

"These results suggested the potential role of CD206 and CD163 in bacteria capture (*vide infra*)". This comment does not make sense and needs to be better justified.

Figure 3c (and other graphs in the manuscript). Y axis needs to start at 0 and statistical test is missing.

"Obviously, the capture capacity of the IL-4-stimulated GM ϕ s was much better than that of unstimulated GM ϕ s for both species (Fig. 3a)." Could this be explained because IL-4-treated cells might have taken more magnetic particles because of a more developed endocytic compartment? Please discuss.

“The formation of bacteria aggregates around the cells was probably due to the unique topological structures of the GM ϕ particles²⁹, as evidenced by the scanning electron microscopy analysis (Supplementary Fig. 10), which dramatically increased the interaction chance between GM ϕ s and bacterial cells.” I do not find this explanation convincing. The cell membrane in Sup Fig 10 looks quite smooth while images in Figure 3a suggest the presence of membrane processes, particularly for *S. aureus*.

Data in Figure 3d needs quantification. The authors should demonstrate not only capture of all species, but also that test if there is preferential uptake of a particular one.

Reagents:

Anti-CD115 antibody is specific against a particular phosphorylated tyrosine. Probably binds intracellular domain.

Supplementary material:

Sup Figs 3, 5, 15, 17, 23: because of only two groups authors should use a T test.

Sup Fig 7: Assay conditions and numbers of repeats need to be included.

Sup Fig 11 needs a positive control to demonstrate PI is working. Also, authors need to mention how representative the images are.

Sup Fig 22: Unclear how MFI is calculated if there are two picks in the plot. In this instance, since only a concentration of toxin and Ab are used, I would suggest plotting % dead cells.

Sup Fig 12: Do the authors compare the level of gelation between macrophages and MCF7 cells?

Sup Figure 14: The authors should include as control an antibody that binds to the surface of macrophages but is not specific for CD206 or SR-Ab to indicate if the effect could be caused just by steric hindrance. See my note about the anti-CD115 Ab recognising an intracellular domain. Treatment of *S. aureus* with Fc fragments needs to be described.

Reviewer #3 (Remarks to the Author):

I have reviewed the revised version and response letters and I think the reviewers have addressed my comments. I recommend publication of this paper in Nature communications.

Detailed Changes of the Manuscript and the Point-by-Point Response to the Editor and Reviewers' Comments

Reviewer #1 (Remarks to the Author):

The authors have positively addressed all my prior comments. The revised manuscript has substantial data to support the novel cell-based construct for bacterial detection. The clinical study further supports the translational viability of the system. The authors are congratulated for a study well done. Some proreading should be performed prior to publication. For instance, line 271 on page 10 "which was significantly than that" should be "which was significantly lower than that". Also Line 289 on page 10 "proofing" should be "proving".

Response: Many thanks for your positive comments, and we have corrected these errors in the revised manuscript.

Reviewer #2 (Remarks to the Author):

The authors have addressed multiple of my queries and the manuscript has been improved. Nevertheless, there are some aspects that require clarification and revision.

Main manuscript

Introduction:

“After treatment with IL-4, M0-type Mø̄s are transformed into M2-type Mø̄s with upregulation of some unique membrane markers such as CD115, CD163, CD206, and CD209”. This section needs a reference. CD209 is also able to recognize bacteria but CD209 is not expressed in mouse cells. As far as I know CD115 is not a classic marker of M2 macrophages.

Response: Many thanks for the helpful suggestions. We have added references in the revised paper (Ref. Shrivastava, R.; Shukla, N., Attributes of alternatively activated (M2) macrophages. *Life Sciences* 2019, 224, 222-231).

CD209a, i.e., dendritic cell-specific ICAM-3-grabbing nonintegrin (DC-SIGN), typically exists on the human DC or macrophage cells, but CD209b known as SIGNR1 is identified on the mouse cells, which is a homologue of human DC-SIGN and may be a M2 phenotype biomarker on the RAW264.7 cells (Please see Reference: *Journal of Cellular Biochemistry*, 2016, 117, 1158–1166; *Int. J. Cancer*, 2017, 141, 1690–1703; *ACS Appl. Mater. Interfaces* 2016, 8, 43, 29310–29322).

Gene	Synonyms	Species	Marker Type	Protein Type	Localiza
ADGRE1	F4/80, EMR1	Hu, Mo	Pan, Tissue	Receptor	Cell Membrane
CCR2	CD192	Hu, Mo	Monocyte, TAM	Receptor	Cell Membrane
CD14		Hu, Mo	Pan, Tissue (cardiac)	Receptor	Cell Membrane
CD68	SCARD1	Hu, Mo	Pan, Monocyte, Tissue (Kupffer, alveolar, interstitial, marginal zone, metaophillic, white pulp)	Receptor	Cell Membrane
CSF1R	CD115	Hu, Mo	Pan, M2, Monocyte, TAM	Receptor	Cell Membrane

As for CD115, it is also recommended as a surface marker for M2-type RAW264.7 cells by suppliers (Please see <https://www.biocompare.com/Editorial-Articles/566347-A-Guide-to-Macrophage-Markers/>). However, we acknowledge that both CSF1R/CD115 and CD209b are not widely accepted markers for M2-type macrophages, so we correct this sentence in the revised manuscript (Please see “*After treatment with IL-4, M0-type Mφs are transformed into M2-type Mφs with upregulation of some unique markers such as CD206, CD163, Arginase-1 (Arg1), and Fizz1 and so on*”).

Results section:

Data in Figure 1b requires isotype controls and the differences between “with and without IL-4” would be clearer if histograms were shown as overlays.

Response: Many thanks for the helpful suggestions, so we test the cells using Rabbit IgG control antibodies (Figure 1b), and no background signal is detected. Of note, all cell samples have been treated using FcR-blocking reagents to inhibit the nonspecific adsorption of antibodies on the macrophages, so the background signal is suppressed. We have also shown the histograms of Figure 1b in the SI (Please see Figure S5) due to the limited space in Figure 1.

Supplementary Figure 5. The effects of IL-4 treatment on (a) TLR2, (b) TLR4, (c) CD206 (MR), and (d) CD163 (SR) expression on the Mø membrane were detected by flow cytometry. (e) Cells treated with Rabbit IgG antibodies (isotype antibody) as a negative control. MFI: mean fluorescence intensity. The error bars represent mean \pm SEM, $n = 3$. Statistical analysis was performed using two-sample t test.

“These results suggested the potential role of CD206 and CD163 in bacteria capture (vide infra) “. This comment does not make sense and needs to be better justified.

Response: Many thanks for the helpful suggestions, so we correct this sentence in the revised manuscript (Please see “*The up-regulation of CD206 and CD163 implied that they could have more chances to capture bacteria compared with TLR receptors*”).

Figure 3c (and other graphs in the manuscript). Y axis needs to start at 0 and statistical test is missing.

Response: Many thanks for the helpful suggestions, so we have corrected the Y axis of figures in the revised manuscript (Please see Figure 3c, 3e and some figures in the SI).

“Obviously, the capture capacity of the IL-4-stimulated GMø was much better than that of unstimulated GMø for both species (Fig. 3a).” Could this be explained because IL-4-treated cells might have taken more magnetic particles because of a more developed endocytic compartment? Please discuss.

Response: (1) ICP-MS results revealed no obvious difference in MNP uptake between IL-4-stimulated Mø and unstimulated Mø. (2) SEM results also verify no noticeable morphology change between the two cells. (3) We perform a capture experiment that uses the unstimulated/IL-4 stimulated GMø without MNP treatment. Again, more bacteria can be captured by IL-4-stimulated GMø. So, the enhanced capture efficiency can't be ascribed to the MNP uptake and the number of endocytic compartments.

(A) MNP uptake and (B) SEM images of unstimulated/IL-4 stimulated GMø without magnetization. Scale bar: 2.5 μ m. The error bars represent mean \pm SEM, n = 3. Statistical analysis was performed using two-sample *t* test. (C) Images of the colonies formed on Luria-Bertani (LB) broth-agar plates captured by IL-4-stimulated GMø with or without magnetization.

“The formation of bacteria aggregates around the cells was probably due to the unique topological structures of the GMø particles²⁹, as evidenced by the scanning electron microscopy analysis (Supplementary Fig. 10), which dramatically increased the interaction chance between GMø and bacterial cells.” I do not find this explanation convincing. The cell membrane in Sup Fig 10 looks quite smooth while images in Figure 3a suggest the presence of membrane processes, particularly for *S aureus*.

Response: Many thanks for the helpful suggestions. We note this problem, and the microstructures on the cell membrane are not enough to induce the bacteria aggregation. So, we remove this description in the revised manuscript, and the exact reason remains unknown.

Data in Figure 3d needs quantification. The authors should demonstrate not only capture of all species, but also that test if there is preferential uptake of a particular one.

Response: Many thanks for the helpful suggestions, so we have provided quantitative results in the revised supporting information (Figure S15). From these results, we uncover that GMøs exhibited considerable capture efficiency for all tested species, which prefer to capture more *P. aeruginosa* compared with other microorganisms.

Supplementary Figure 15. Colony statistics of different microorganisms captured by GMøs using a plate assay. The error bars represent mean \pm SEM, n = 3. Statistical analysis was performed using one-way analysis of variance (ANOVA).

Reagents:

Anti-CD115 antibody is specific against a particular phosphorylated tyrosine. Probably binds intracellular domain.

Response: Many thanks for the helpful suggestions, so we use another antibody (anti-F4/80 antibody) to avoid this problem in the revised manuscript.

Supplementary material:

Sup Figs 3, 5, 15, 17, 23: because of only two groups authors should use a T test.

Response: Many thanks for the helpful suggestions, so we have used *t* test in the revised Figures.

Sup Fig 7: Assay conditions and numbers of repeats need to be included.

Response: Many thanks again, so we have revised Figure S7 and added more description about assay conditions and numbers of repeats (Please see “**Supplementary Figure 7.** Cell viability of Møs before and after gelation. Raw 264.7 cells were cultured on a 96-well plate (1×10^4 cells/well). After 12 h, cells without/with gelation were treated with the 10% (v/v) CCK-8 reagent in DMEM medium for 0.5 h at 37 °C, and the absorbance at 450 nm was recorded by a microplate reader. The error bars represent mean \pm SEM, n = 3.”).

Sup Fig 11 needs a positive control to demonstrate PI is working. Also, authors need to mention how representative the images are.

Response: Many thanks for the helpful suggestions, so we have revised Figure S11 and provided representative pictures with more bacteria. From the results, the captured bacteria in the positive control group killed by heat can be stained by PI, thus

demonstrating its effectiveness.

Supplementary Figure 10. Representative fluorescence images of isolated *E. coli* by GMøS after staining with DAPI and PI dyes. DAPI fluorescence shows the location of bacteria, and PI fluorescence indicates the dead bacteria. There is no obvious difference between normal *E. coli* and captured *E. coli*, and no PI fluorescence is observed, thus verifying that the captured bacteria are still alive. Positive control shows that the captured *E. coli* killed by heat can be stained by PI. Scale bar: 20 μm .

Sup Fig 22: Unclear how MFI is calculated if there are two peaks in the plot. In this instance, since only a concentration of toxin and Ab are used, I would suggest plotting % dead cells.

Response: Many thanks for the helpful suggestions, so we have corrected Figure S22 in the revised SI.

Supplementary Figure 22. Flow cytometry analysis of GMøS with different treatments and subsequently stained by PI molecules. GMøS treated with Hl α , Hl α +anti-SR antibody, and Hl α +anti-Hl α antibody for 0.5 h. GMøS incubated with PI was used as negative control. [Hl α] = 10 nM, [antibody] = 3.3 $\mu\text{g mL}^{-1}$.

Sup Fig 12: Did the authors compare the level of gelation between macrophages and MCF7 cells?

Response: We use the same procedures to prepare gelated MCF-7 cells and also confirm the gelation of MCF-7 cells by pure water test and fluorescence microscopy. We just test the interaction between MCF-7 cells and bacteria, so we do not use additional methods to evaluate the gelation level of MCF-7 cells.

Sup Figure 14: The authors should include as control an antibody that binds to the surface of macrophages but is not specific for CD206 or SR-Ab to indicate if the effect could be caused just by steric hindrance. See my note about the anti-CD115 Ab recognizing an intracellular domain. Treatment of *S. aureus* with Fc fragments needs to be described.

Response: Many thanks for the helpful suggestions. In our revised manuscript, we use another antibody (antiF4/80 antibody) to avoid this problem. F4/80 is a surface marker of macrophages, which is not directly related to bacteria recognition. Again, the bacteria capture is not disturbed by adding antiF4/80 antibodies.

Treatment of *S. aureus* with Fc fragments has been described in the revised manuscript (Please see “*To perform the antibody-blocking assay, GMØs (1×10^5 cells) were treated with FcR Blocking Reagent for 40 min. Then, the potential bacteria-binding receptors were blocked by corresponding antibodies (1:300) for 1 h with continuous shaking (350 rpm). In parallel, S. aureus was incubated with mouse Fc fragments (2 mg mL^{-1}) to block protein A on the bacterial surface.*” Page 20, section of Bacteria capture).

Reviewer #3 (Remarks to the Author):

I have reviewed the revised version and response letters and I think the reviewers have addressed my comments. I recommend publication of this paper in Nature communications.

Response: Many thanks for your positive comments.

REVIEWERS' COMMENTS

Reviewer #2 (Remarks to the Author):

The authors have addressed my queries. Just minor edits to improve clarity.

Line 168: Do they mean that "upregulation of MR and CD163 would increase the pathogen-binding capacity of the cells"?

Blocking CD206 and CD163 significantly reduces number of bacteria capture but the authors should acknowledge that there is substantial activity remaining so other receptors could be involved.

Detailed Changes of the Manuscript and the Point-by-Point Response to the Editor and Reviewers' Comments

Reviewer #2 (Remarks to the Author):

The authors have addressed my queries. Just minor edits to improve clarity. Line 168: Do they mean that "upregulation of MR and CD163 would increase the pathogen-binding capacity of the cells"?

Blocking CD206 and CD163 significantly reduces number of bacteria capture but the authors should acknowledge that there is substantial activity remaining so other receptors could be involved.

Response: Yes, we want to show that the upregulation of CD206 and CD163 would increase the pathogen-binding capacity of the cells (Please see "In parallel, these cells are stimulated by a cytokine (e.g., interleukin-4, IL-4) to induce the upregulation of pathogen-binding receptors (e.g., MR and CD163).", Page 4 in the main text).

Of course, the use of blocking antibodies can't completely inhibit the binding of bacteria to macrophages, so we acknowledge that substantial activity remaining so other receptors could be involved (Please see "Of note, the use of blocking antibodies could not completely inhibit the binding of bacteria to GMØs, so we acknowledged that substantial activity remaining so other receptors could be involved.", Page 6 in the main text).